# A shear stress micromodel of urinary tract infection by the *Escherichia coli* producing Dr adhesin

**Beata Zalewska-Piątek**[1], **Marcin Olszewski**[1], **Tomasz Lipniacki**[2], **Sławomir Błoński**[2], **Miłosz Wieczór**[3], **Piotr Bruździak**[3], **Anna Skwarska**[4], **Bogdan Nowicki**[5], **Stella Nowicki**[5], **Rafał Piątek**[1]*

**1** Department of Molecular Microbiology and Biotechnology, Gdańsk University of Technology, Gdańsk, Poland, **2** Department of Biosystems and Soft Matter, Institute of Fundamental Technological Research, Polish Academy of Sciences, Warsaw, Poland, **3** Department of Physical Chemistry, Gdańsk University of Technology, Gdańsk, Poland, **4** Department of Oncology, University of Oxford, Oxford, United Kingdom, **5** Nowicki Institute for Woman's Health Research, Brentwood, Tennessee, United States of America

* rafpiate@pg.edu.pl

**Data Availability Statement:** All relevant data are within the manuscript and its Supporting Information files.

## Abstract

In this study, we established a dynamic micromodel of urinary tract infection to analyze the impact of UT-segment-specific urinary outflow on the persistence of *E. coli* colonization. We found that the adherence of Dr+ *E. coli* to bladder T24 transitional cells and type IV collagen is maximal at lowest shear stress and is reduced by any increase in flow velocity. The analyzed adherence was effective in the whole spectrum of physiological shear stress and was almost irreversible over the entire range of generated shear force. Once Dr+ *E. coli* bound to host cells or collagen, they did not detach even in the presence of elevated shear stress or of chloramphenicol, a competitive inhibitor of binding. Investigating the role of epithelial surface architecture, we showed that the presence of budding cells–a model microarchitectural obstacle–promotes colonization of the urinary tract by *E. coli*. We report a previously undescribed phenomenon of epithelial cell "rolling-shedding" colonization, in which the detached epithelial cells reattach to the underlying cell line through a layer of adherent Dr+ *E. coli*. This rolling-shedding colonization progressed continuously due to "refilling" induced by the flow-perturbing obstacle. The shear stress of fluid containing free-floating bacteria fueled the rolling, while providing an uninterrupted supply of new bacteria to be trapped by the rolling cell. The progressive rolling allows for transfer of briefly attached bacteria onto the underlying monolayer in a repeating cascading event.

## Author summary

Uropathogenic *E. coli* (UPEC) equipped with Dr fimbriae are associated with recurrent and chronic urinary tract infections (UTIs). The fimbriae assembled by the chaperone-usher pathway provide strong host-specific adherence which is, however, strongly modulated by the dynamically changing urine flow in the urinary tract (UT). In this paper, we use a dynamic *in vitro* micro-model of UTI to analyze the UT segment-specific impact of

**Funding:** This work was supported by Polish National Science Centre Grant 2014/13/B/NZ6/00257 to R. P. The funders had no role in study design, data collection and analysis, decision to publish, or preparation of the manuscript.

**Competing interests:** The authors have declared that no competing interests exist.

urinary outflow on the persistence and spread of Dr+ *E. coli* during host colonization. We conclude that the adhesive envelope formed by Dr fimbriae promotes strong and irreversible multivalent adherence of Dr+ *E. coli* to host receptors under flow conditions. We also observed that budding host cells–a model of any form of epithelial roughness, including carcinogenesis or physical injuries–facilitate the adherence of bacteria at flow conditions typically found in the UT, and our numerical simulations provided a mechanistic explanation for this effect. Finally, we combined the results to propose a rolling-shedding-refilling colonization model that shows how the wash off of detached colonized host cells may provoke a massive spread of UPEC. Our findings shed new light on UTI development and may be instrumental in the development of novel therapeutics.

## Introduction

Urinary tract infections (UTIs) are among the most common nosocomial and community-acquired bacterial infections affecting up to 150 million people worldwide each year [1]. 65–85% of UTIs are caused by uropathogenic *E. coli* strains (UPECs) that reside in the intestine and colonize the periurethral area, ascending stepwise to the bladder, ureters, and kidneys [2,3]. The ascending colonization is facilitated by *E. coli* adhesins that allow pathogens to anchor to specific receptors expressed by uroepithelial lining of the urinary tract, evading elimination by urine flow.

The adhesive structures of *E. coli* are assembled by the highly conserved chaperone-usher secretion pathway (CUP), with some UPEC clinical strains containing up to 16 different CUP adhesive structures [4–6]. Among the adhesins, the most thoroughly characterized are the cystitis-associated (type 1) and pyelonephritis-associated (P) pili, encoded by the *fim* or *pap* operon, respectively. Both are monoadhesins: heteropolymers with a complex subunit composition tipped with a single adhesin molecule, FimH or PapG [7]. Over 90% of UPEC strains produce type 1 pili that mediate mannose-specific adhesion to the bladder epithelium by binding to mannosyl residues *via* the lectin domain of FimH. The third most common group of UPEC adhesins–members of the Dr/Afa family–is often associated with cystitis in children, as well as pyelonephritis and recurrent treatment-resistant UTIs in young and pregnant women [8]. In contrast to monodhesive pili, these Dr adhesins are polyadhesins composed of hundreds of adhesive DraE subunits capped with a single DraD subunit [9–12]. In addition, DraE homopolymers bind to a range of cellular receptors: the human decay-accelerating factor (DAF, CD55), members of the carcinoembryonic antigen-related cell adhesion molecules (CEACAM) family such as CEACAM1 (CD66e) or CEACAM6 (CD66c) and type IV collagen [13–16]. Binding of Dr adhesins to DAF, CEACAM receptors and type IV collagen is competitively inhibited by chloramphenicol [15–19].

The occurrence of specific interaction between UPEC adhesins and surface located host receptors is crucial for bacterial adherence. This stage of pathogenesis is, however, strongly modulated by the dynamically varying shear stress associated with urine flow in the urinary tract. From the glomerulus through the renal tubules, renal pelvis and ureters, the stress due to urine flow is amplified by the peristalsis of ureters. This peristalsis creates high velocity ureteral jets to the bladder, and further contractions of the bladder induce rapid urethral flow with urination. The high velocity outflow is controlled by a system of sphincters, which cooperate in the removal of planktonic bacteria and bodily waste–including infected, unhealthy, and/or detached epithelial cells. The magnitude of flow-induced stress ranges from 0.017 pN $\mu m^{-2}$ in the proximal renal tubule to 0.3–0.5 pN $\mu m^{-2}$ in the urethra [20]. The shear stress then affects

bacterial adherence in a manner dependent on the mechanism of adhesin-receptor binding. Two such broadly defined mechanisms are considered in the literature: slip-bonds, characterized by an exponential decay of binding lifetimes with increasing tensile stress, and catch-bonds, exhibiting maximum stability in high stress conditions. As a result, bacterial adherence mediated by slip-bonds is the most stable in no-flow conditions, while bacteria that use catch-bonds adhere optimally at higher flow rates [20–22].

Importantly, *E. coli* strains producing type 1 pili have been shown to adhere to host cells through the catch-bond mechanism, effected by the peculiar structure of the FimH adhesin and its mode of interaction with the mannosyl residues [23–24]. This means that under low shear stress, bacteria bind to cell surface frequently but transiently, and may roll on the receptor-presenting host cells, spreading to other body niches during episodes of low flow or in static conditions. In contrast, at higher shear stress the bacteria stick firmly to the surface, which prevents them from being washed off the tissue. The catch-bond mechanism also effectively counteracts the host defense mechanism based on the induction of intense exfoliation and elimination of the bladder epithelium cells infected with bacteria. This is because the bacteria that adhere to the exfoliated epithelium are no longer subjected to shear stress and dissociate, passively raised by the flow of urine to colonize other cells and facilitate further progression of UTI [25–28].

In turn, the mechanism of adherence of Dr fimbriae remains largely unexplored, and the adhesive structures exhibit drastically different morphology and dynamic properties compared to type 1 and P pili. Dr fimbriae interlace extensively, forming an amorphous capsule surrounding the bacteria, and this polyadhesive envelope contains DraE subunits capable of interacting with host receptors in its entire volume [29]. It is therefore natural to ask whether the adherence of Dr-positive (Dr+) bacteria follows the catch- or slip-bond model. Furthermore, the Dr/Afa producing UPECs are often associated with chronic or recurrent UTIs [8]. This raises another question: to what extent the observed features of UTIs are determined by the mechanism of Dr adhesion under shear stress in the urinary tract?

Despite extensive research on the pathophysiological effects of shear stress, there are no systematic studies on the capacity of urine flow to detach Dr+ *E. coli* in different parts of the urinary tract. More importantly, no past studies addressed the effect of shear stress on the elimination of infected, unhealthy, or dead epithelial cells and their eventual impact on UTI progression. In this paper, we approach these questions using a dynamic micromodel of urinary tract infection to analyze the UT segment-specific impact of urinary outflow on the persistence and spread of *E. coli* colonization. We conclude that the adherence of Dr+ *E. coli* strains to bladder T24 transitional cells and type IV collagen is maximal at lowest shear stress, consistently with the slip-bond model. We show how any structural distortions of cell surface enhance bacterial adherence through alteration of flow profiles. We also describe a previously unknown phenomenon of epithelial cell "rolling-shedding" colonization: a transfer of bacteria from a rolling urothelial cell onto the underlying cell layer in a repetitive cascading event that promotes progressive colonization instead of elimination of the detached urothelium.

## Results

### Adherence of Dr+ *E. coli* to bladder T24 transitional cell line is only dependent on interactions with DAF receptor

As model host cells for studying the adherence of Dr+ *E. coli*, we used the transitional bladder urothelial T24 cell line. Urothelium covers the lumen of ureters, bladder and part of urethra, and hence is frequently employed in experiments with UPEC strains [30–32]. We also confirmed that our recombinant Dr+ strain produced plasmid-encoded Dr fimbriae at a level

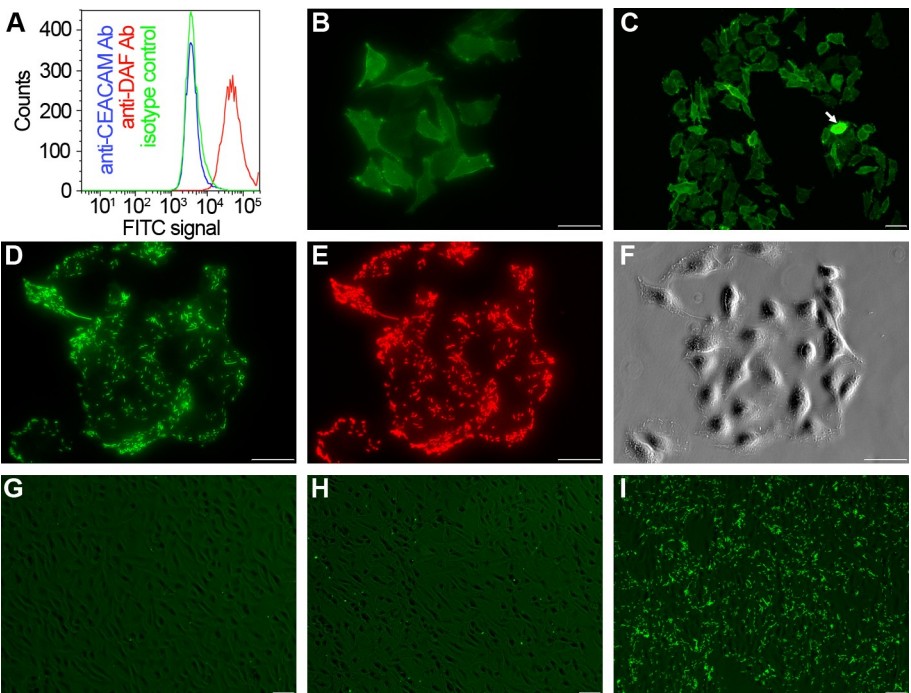

**Fig 1. Dr+ *E. coli* adhere to bladder T24 cells through interaction with DAF.** (A) Surface expression of DAF and CEACAM receptors in T24 bladder carcinoma cells. FACS histograms of a representative labeling of T24 cells with antibodies against human DAF (red curve) or CEACAM (blue curve) conjugated with FITC are shown in comparison to labeling with the isotype control antibody (green line). T24 cells constitutively express DAF, but not CEA receptor. Data is representative of three repeats. (B,C) Immunofluorescence analysis of T24 cells using polyclonal anti-DAF antibodies. The arrowhead marks a single cell that overexpresses the DAF receptor. (D-F) T24 cells incubated with Dr+ *E. coli* in static conditions. Unbound bacteria were washed off, and bound ones were visualized via endogenous GFP fluorescence (D) or immunolabeling with anti-Dr antibodies (E). Panel (F) was generated using phase-contrast microscopy. (G-H) T24 cells incubated with (H) or without (G,I) polyclonal anti-DAF antibodies and washed with Dr- (G) or Dr+ (H,I) *E. coli*. Unbound bacteria were washed off, and remaining ones were visualized using GFP fluorescence. Pictures (B-I) are representative of four independent replicates. Scale bars correspond to 50 μm.

comparable with the clinical prototypic IH11128 UPEC strain (see S1 Fig), making it a suitable model of clinical UTIs.

We used flow cytometry to verify the presence of receptors recognized by Dr fimbriae on T24 cells. As shown in Fig 1A, measurements with anti-DAF and anti-CEACAM antibodies confirmed the presence of DAF and absence of carcinoembrionic antigens. An immunofluorescence assay with anti-DAF antibodies indicated that T24 cells express DAF homogeneously, with a noticeable overexpression observed only in ca. 0.5% of the population (Fig 1B and 1C). When these DAF-presenting T24 cells were incubated with Dr+ *E. coli* in static conditions, the bacteria produced a diffuse pattern of adherence characteristic for Dr/Afa producing UPEC strains (Fig 1D–1F) [33,34]. Pre-incubation of T24 cells with anti-DAF antibodies resulted in 97 ± 8% reduction of adherence in comparison to untreated cells. The negative control (Dr-) strain adhered to T24 cells at 2 ± 1% of the value observed for the Dr+ strain (Fig 1G–1I). This data shows that the adherence of Dr+ bacteria to T24 cells is almost exclusively mediated by specific interactions between Dr fimbriae and host DAF receptors.

### Shear stress reduces the adherence of Dr+ *E. coli* to host cells by lowering the success rate of productive initial attachment

To study the adherence of Dr+ *E. coli* in real time, bacterial suspension was passed through a parallel plate flow chamber over a confluent layer of T24 cells. Attachment was monitored

using GFP fluorescence at different flow rates corresponding to shear stress ranging from 0.01 to 1.15 pN μm$^{-2}$. To directly compare our data with that published for type 1-piliated *E. coli* for which the catch bond mechanism is well established, we used identical flow parameters in our experiments [35]. Fig 2A shows representative bacterial accumulation curves recorded at different values of shear stress, and S1 Video illustrates the process of bacterial accumulation at four different flow rates. At each value of shear stress, the number of attached bacteria increased linearly with time, indicating that the host receptors are not saturating with Dr fimbriae. At the same time, adherence of the Dr- strain to T24 cells ranges from 1.7 ± 1% to 5 ±6% of the reference values in adhesion-competent systems at shear stresses of 0.01 and 1.15 pN μm$^{-2}$, respectively (Fig 2B and 2C, S2 Video).

As seen in Fig 2A–2C, increasing shear stress reduced the number of accumulated bacteria through reduction in attachment rate, defined as the number of attachment events per minute observed in the field of view (Fig 2B). Indeed, an increase of shear stress from 0.01 to 0.28 pN μm$^{-2}$ more than halved the attachment rate from 67 ± 5 to 31 ± 4 bacteria per minute. At the maximal shear stress of 1.15 pN μm$^{-2}$, the observed attachment rate was as low as 6 ± 2 bacteria per minute. Thanks to infrequent detachment events, in shear stress ranging from 0.01 to 0.58 pN μm$^{-2}$ the efficiency of adherence–defined as the ratio of accumulation rate to attachment rate–is high and equal to 72–80 ± 17% (inset in Fig 2B), only to drop to 39 ± 19% at the highest shear stress.

Fig 2C reports the statistics of adherence after 12 minutes of flow (the time at which no disturbances of the cell line was observed), including the total number of attachment events, number of fast detachments (within 1 second from initial binding), number of slow detachments (after more than 1 second from initial binding), and the final bacteria count at the 12 minutes mark. At a shear stress of 0.01 pN μm$^{-2}$, a total of 799 ± 57 attachment events could be subdivided into 130 ± 23 (16%) fast detachments, 648 ± 51 (81%) stably adhered bacteria and only 22 ± 3 (3%) slow detachments. As the shear stress increased to 0.28 pN μm$^{-2}$, the fraction of total detachment events remained stable at ca. 20%, but the proportion of fast and slow detachments changed gradually in favor of the latter (Fig 2C). Eventually, at high values of shear stress (0.58 to 1.15 pN μm$^{-2}$) fast detachment events were no longer observed. Overall, this indicates that shear stress both reduces the number of initial adherence events and facilitates the immediate detachment of those that bind ineffectively: at high shear stress, short-lived adherence events are effectively prevented by drag force.

## Cell surface-bound Dr+ *E. coli* are resistant to detachment by high shear stress

The accumulation experiments suggest that after successful initial adherence to T24 host cells, most bacteria remain stably attached over long time periods. To check the strength of bacterial adhesion under varying shear stress, we first passed Dr+ *E. coli* through the flow chamber at low shear stress (0.01 pN μm$^{-2}$) to accumulate, and then subjected the cell-bound bacteria to shear stress increasing in discrete steps from 0.06 to 18.46 pN μm$^{-2}$. As seen in Fig 2D and S3 Video, at up to 4.62 pN μm$^{-2}$ 96 ± 20% of initially bound bacteria remained stably adhered. The subsequent increase of shear stress to 9.23 and 18.46 pN μm$^{-2}$ increased the fraction of detached bacteria to ca. 20 and 30%. In the two highest flow rates, the observed reduction in bacteria count did not result from individual bacterial detachment events but rather the detachment of individual host cells covered in bacteria due to cell layer instability at high shear stress (cf. S3 Video). S2A Fig shows the statistics of bacterial detachment at each value of shear stress employed, in the range from 0.06 to 4.62 pN μm$^{-2}$.

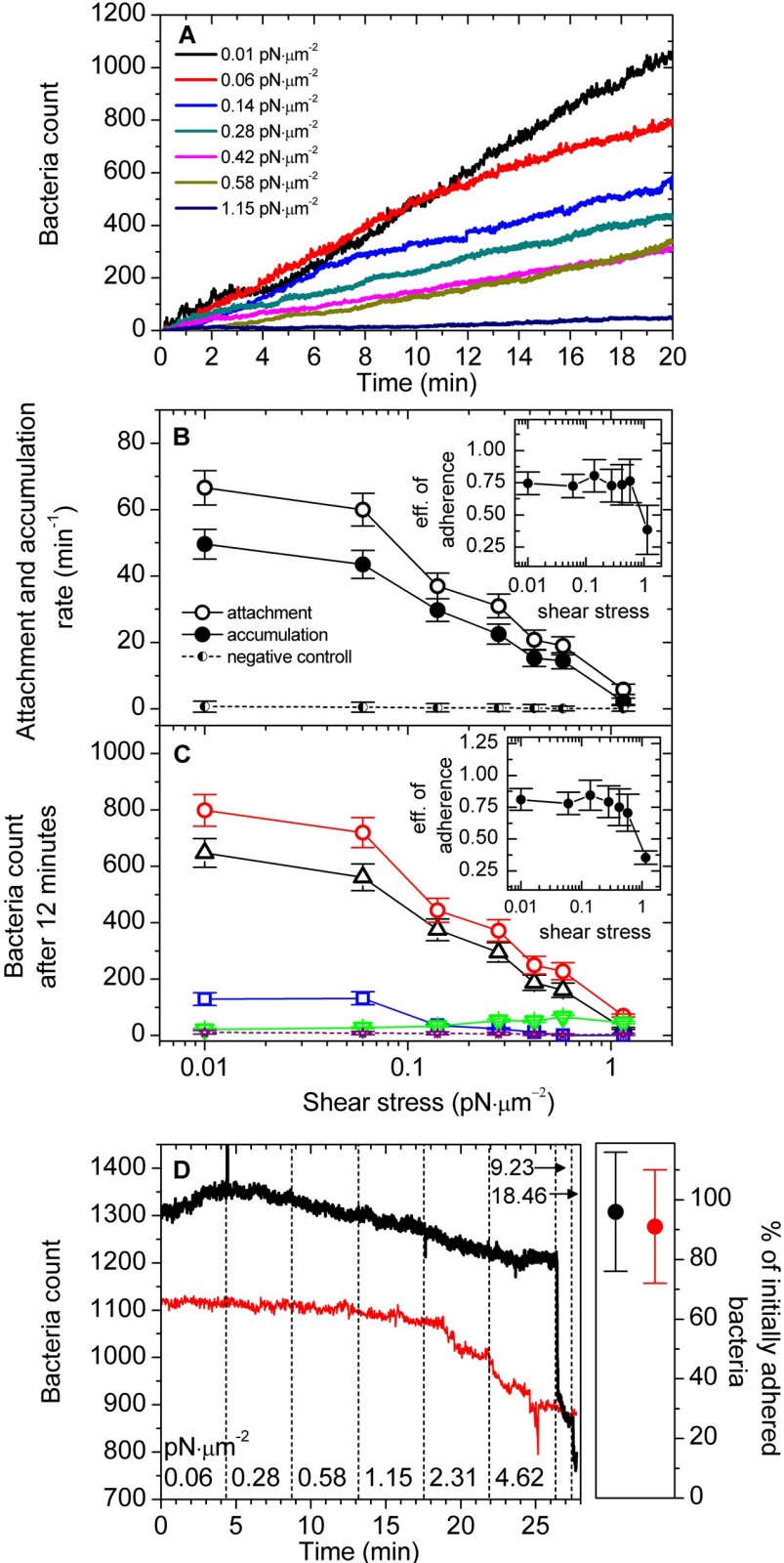

**Fig 2. Adherence of Dr+ *E. coli* to T24 cells is inhibited by increasing shear stress.** (A) Accumulation of Dr+ *E. coli* over 20 minutes of flow at different values of shear stress. One representative dataset out of 12 replicates is shown. (B) Attachment rate (open circles) calculated as the total count of binding events divided by total time, and accumulation

rate (full circles) calculated from the accumulation curves in panel A. The dashed line represents accumulation rate for Dr- (non-fimbriated) *E. coli*. Inset: adherence efficiency of Dr+ bacteria at individual shear stresses, defined as the ratio of accumulation to attachment rates. (C) Classification of attachment events according to the ultimate fate of the bound bacterial cell: red–total attachment events (recorded at one-frame resolution over 12 minutes); blue–detachment events within 1 second from the initial binding; green points–detachment events more than one second from the initial binding; black–persistently bound bacteria. The dashed line represents the number of accumulated Dr- *E. coli* after 12 minutes. Inset: adherence efficiency of Dr+ bacteria, here defined as the ratio of bacterial accumulation after 12 minutes to the attachment events count during the 12 minutes. The error bars in (B) and (C) give 95% confidence intervals of 12 replicates. (D) Black line–results of the detachment experiment. ca. 1350 initially bound bacteria were tracked for eventual detachment by the increasing flow of medium, generating shear stress ranging from 0.06 to 18.46 pN $\mu m^{-2}$ over 28 minutes. The flow was increased stepwise without pausing between subsequent flow values, as denoted in the figure. Red line–representative result of the detachment experiment with 1150 bacteria initially bound. Each flow step was separated from the next by a 2-minute pause (not included in the time scale), totaling 42 minutes. One representative experiment out of 12 replicates is shown. The error bars give 95% confidence intervals of 12 replicates, and correspond to a fraction of initially bound bacteria that remain bound at the end of flow (at a shear stress of 4.62 pN $\mu m^{-2}$). S2 Fig shows statistics of bacteria that remain bound at the end of each flow step relative to the beginning of the same step, at flows ranging from 0.06 to 4.62 pN $\mu m^{-2}$.

We also noted that in the urinary tract, urine flow is not continuous but includes alternating stages of peristaltic waves, ureteral urine jets or filling and voiding of the bladder. To mimic this processes, we repeated the detachment experiments with consecutive stages of medium flow separated by 2-minute periods of stopped flow (Fig 2D). The results were almost identical as in the continuous flow experiment with no bacterial detachment observed in periods of no flow, i.e. when the bacteria were not subjected to drag force. As in the continuous flow experiment, 91 ± 19% of initially bound bacteria remain stably adhered with shear stress gradient up to 4.62 pN $\mu m^{-2}$. Eventually, cell layer instability could be observed for shear stress of 4.62 pN $\mu m^{-2}$, as periods of stopped flow extended the duration of the experiment. S2B Fig shows the statistics of bacterial detachment at each value of shear stress employed, in the range from 0.06 to 4.62 pN $\mu m^{-2}$.

## Binding of chloramphenicol to Dr fimbriae reduces the attachment rate to host cells

The prevalence of recurrent UTIs motivates the search for small-molecule binding inhibitors that block the attachment or remove *E. coli* from the urinary tract [36]. One such lead compound is chloramphenicol (Cm), a popular antibiotic that was shown to compete with receptor binding by DraE in stationary systems [15,37]. To test whether these findings hold under flow conditions, we analyzed the adhesive properties of Dr+ *E. coli* pre-incubated in 50, 100 and 200 μM Cm, using a shear stress that previously yielded efficient adherence (0.01 pN $\mu m^{-2}$). As seen in Fig 3A–3C, 50 μM Cm yielded accumulation curves, attachment and accumulation rates as well as cumulative counts after 12 minutes almost identical to Cm-free experiments. However, 100 μM Cm reduced attachment and accumulation rates by 34 ± 11% and 71 ± 15%, respectively, with a corresponding drop in attachment event and cumulative counts after 12 minutes (Fig 3B and 3C), while 200 μM Cm almost totally prevented bacterial adherence. The observed adherence efficiency (the ratio of accumulation rate to the attachment rate) dropped from 70 ± 9% in absence of Cm to 66 ± 8%, 31 ± 6% and 8 ± 7% as Cm concentrations of 50, 100 and 200 μM were used (Fig 3B and 3C, insets), mainly due to increase in the number of fast detachment events (Fig 3C). In other words, a partial saturation of binding sites on Dr fimbriae with 100 μM Cm increases the number of bacteria that bind only transiently to host cells and detach rapidly due to drag force. This strongly indicates that the limiting step in adherence of Dr+ bacteria to host cells is the formation of multivalent contacts between the DraE adhesins and the DAF receptors: once the bacterium remains bound long enough to form many DraE-DAF bonds, it will remain stably attached. This rapid multivalent binding also strongly

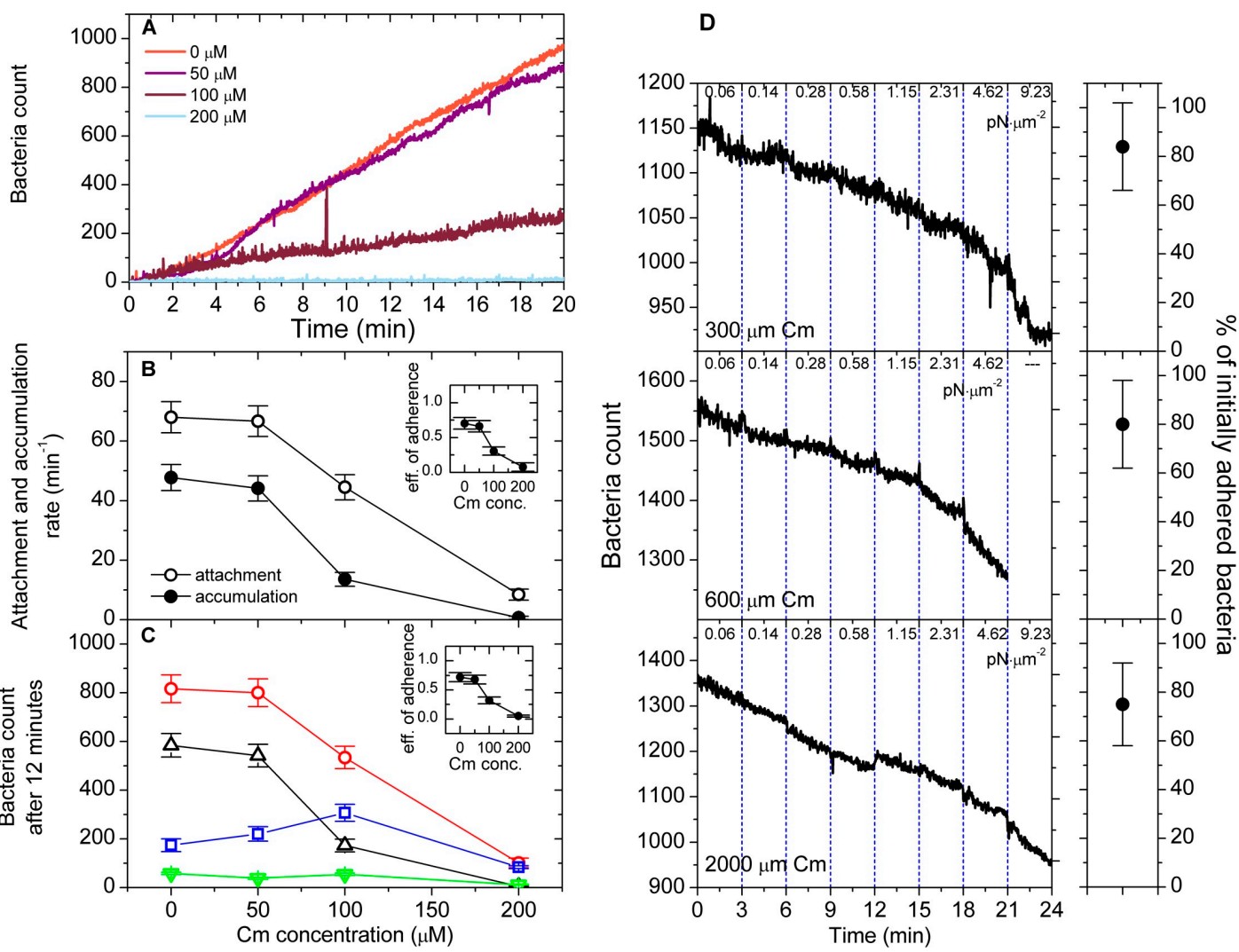

**Fig 3. Chloramphenicol blocks initial bacterial adherence and does not detach pre-bound bacteria.** (A) Accumulation curves of Dr+ *E. coli* pre-incubated with Cm at the indicated concentrations at shear stress of 0.01 pN μm$^{-2}$. Real-time accumulation was monitored using GFP fluorescence. One representative experiment out of 12 replicates is shown. (B) Attachment (open circles) and accumulation (full circles) rates of bacteria pre-incubated with Cm, calculated identically as in Fig 2. Inset: adherence efficiency as a function of Cm concentration. (C) Classification of attachment events according to the fate of the bacterial cell pre-incubated with Cm: red–total attachment events (recorded at one-frame resolution over 12 minutes); blue–detachment events within 1 second from the initial binding; green points–detachment events more than one second from the initial binding; black–persistently bound bacteria. Inset: adherence efficiency as a function of Cm concentration The error bars in (B) and (C) give 95% confidence intervals of 12 replicates (D) Results of the detachment experiments performed at three different Cm concentrations. Ca. 1100–1600 initially accumulated bacteria were tracked for eventual detachment during flow of medium supplemented with Cm at a given concentration. The flow was increased stepwise, generating shear stress ranging from 0.06 to 9.23 pN μm$^{-2}$ as denoted in the figure. For each Cm concentration, one representative experiment out of 12 replicates is shown. The error bars give 95% confidence intervals of 12 replicates, and correspond to a fraction of initially bound bacteria that remain bound at the end of the flow step (at a shear stress of 4.62 pN μm$^{-2}$). S3 Fig shows statistics of bacteria that remain bound at the end of each flow step relative to the beginning of the same step, at flows ranging from 0.06 to 4.62 pN μm$^{-2}$.

relies on the number of available functional DraE subunits, as only a twofold increase in Cm concentration resulted in a 70% reduction of accumulation rate.

## Chloramphenicol does not accelerate pre-bound Dr+ *E. coli* detachment from host cells under shear stress

To evaluate the effect of Cm on the detachment of pre-bound Dr+ *E. coli* from T24 cell lines, Cm-free bacterial suspension was first washed through the flow chamber at a shear stress of

0.01 pN µm$^{-2}$ to establish stable adherence, after which the flow was switched to a medium containing 300, 600 and 2000 µM Cm. At each Cm concentration, the shear stress was gradually increased from 0.06 to 9.23 pN µm$^{-2}$, and bacteria count was monitored. For shear stress exceeding 4.62 pN µm$^{-2}$, however, results cannot be interpreted properly as cell lines began to lose integrity. At this limiting value of stress, the fraction of bacteria that remained bound was equal to 84 ± 18%, 81 ± 18% and 75 ± 17% at Cm concentration of 300, 600 and 2000 µM, respectively (Fig 3D and S4 Video, S3 Fig shows statistics for each value of shear stress employed). This implies that contrary to the findings above, Cm added after bacterial attachment cannot effectively compete for binding with DAF receptors already bound by DraE adhesins, likely due to a large number of existing DraE-DAF contacts that would have to be simultaneously disrupted to release the bound bacteria. A similar effect was observed for adherence of type 1-piliated *E. coli* to immobilized mannose receptors in flow medium containing free mannose as a competitive inhibitor. When bacteria adhered strongly at high shear stress, soluble mannose at high concentration was not enable to detach bacteria. Additionally, when bacteria started to roll at moderate shear stress, soluble mannose facilitated this process and accelerated bacterial spreading [26]. These data together indicate the need to rethink the usage of adherence inhibitors in UTI therapy.

## Adherence of Dr+ *E. coli* to type IV collagen closely resembles adherence to T24 cells

As a basement membrane component, type IV collagen is located below the uroepithelial layer, and becomes accessible as an attachment target following epithelial desquamation, injury due to e.g. shear stress or urologic procedures, or bacterial deep tissue invasion. Respectively, we tested the influence of shear stress on the adherence of Dr+ *E. coli* to human type IV collagen, using flow-chambers coated with three concentrations of collagen– 20, 2 and 0.2 µg ml$^{-1}$ –to estimate the effect of receptor concentration on binding efficiency. As shown in Fig 4A, at shear stresses ranging from 0.01 to 1.15 pN µm$^{-2}$ the number of accumulated bacteria increased almost linearly with time for all three collagen concentrations, again indicating lack of a saturation effect. The observed reduction in bacteria accumulation with increasing shear stress was also consistent with the corresponding result for T24 cells, suggesting the slip-bond model for the adherence of Dr+ bacteria to type IV collagen. While for collagen concentrations of 20 and 2 µg ml$^{-1}$ bacterial accumulation was almost identical at the lowest shear stress (Fig 4A and 4B and S5 Video), bacteria responded differently to collagen concentration at higher flows. A shear stress of 0.06 pN µm$^{-2}$ reduced the bacterial accumulation by 64 ± 13% at the intermediate, but only by 20 ± 13% at the highest collagen concentration (Fig 4A and 4B and S6 Video). At the lowest collagen concentration, bacterial accumulation was already reduced 88 ± 13% with respect to intermediate concentrations at the lowest shear stress of 0.01 pN µm$^{-2}$, and was nearly undetectable at higher flows (Fig 4A and 4B). The adherence of both Dr- strain to type IV collagen and Dr+ strain to the flow chamber incubated with the coating buffer (i.e., in absence of collagen) was unproductive, amounting to <1% of the reference values in adhesion-competent systems (Fig 4B and S2 Video).

Similarly to interaction with DAF receptors, the binding of Dr fimbriae to type IV collagen is blocked by Cm molecules. This permits us to examine the influence of reduction of DraE receptor binding sites on bacterial adherence to collagen (Fig 4D and 4E). We used identical experiment parameters as in the case of Cm inhibition of Dr+ bacteria adherence to T24 cells. The obtained results were very similar to that with a confluent cell line (Fig 3A). Bacteria preincubated with 50 µM Cm yielded accumulation curves almost identical to Cm-free experiments. Incubation of bacteria with 100 and 200 µM Cm resulted in 55 ± 9% and 99 ± 10%

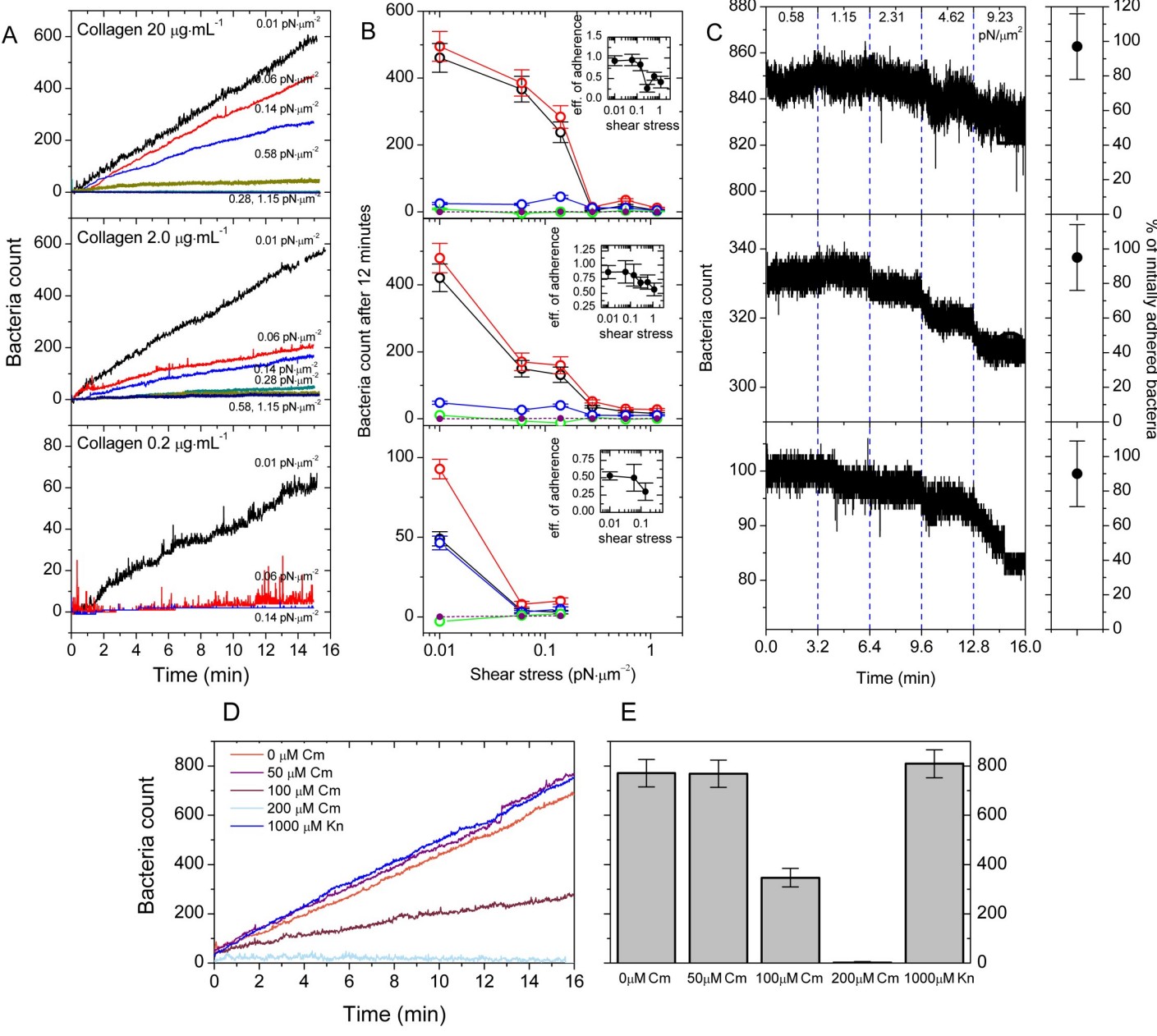

**Fig 4. Shear stress inhibits adherence of Dr+ bacteria to type IV collagen by reducing success rate of initial binding.** (A) Accumulation of Dr+ *E. coli* washed through the flow chamber coated with 20, 2 or 0.2 μg ml⁻¹ human type IV collagen, at shear stress ranging from 0.01 to 1.15 pN μm⁻². Real-time accumulation was monitored using GFP fluorescence. One representative experiment out of 12 replicates is shown. (B) Classification of attachment events according to the ultimate fate of the bacterial cell bound to type IV collagen: red–total attachment events (recorded at one-frame resolution over 12 minutes); blue–detachment events within 1 second from the initial binding; green points–detachment events more than one second from the initial binding; black–persistently bound bacteria. The dashed line represents the number of Dr- *E. coli* accumulated after 12 minutes of flow. Inset: adherence efficiency, defined as the ratio of bacterial accumulation after 12 minutes to the attachment events count during the 12 minutes. The error bars give 95% confidence intervals of 12 replicates. (C) Results of the detachment experiment using type IV collagen. Ca. 100–850 initially accumulated bacteria were individually tracked in stepwise increasing shear stress, ranging from 0.58 to 9.23 pN μm⁻² over 16 minutes, as shown in the figure. For each collagen concentration, one representative experiment out of 12 replicates is shown. Collagen concentrations in panels (B) and (C) are as indicated in panel (A). The error bars give 95% confidence intervals of 12 replicates, and correspond to a fraction of initially bound bacteria that remain bound at the end of flow (at a shear stress of 9.23 pN μm⁻². S4 Fig shows statistics of bacteria that remain bound at the end of each flow step relative to the beginning of the same step. (D) Accumulation of Dr+ *E. coli* pre-incubated with Cm or Kn washed through the flow chamber coated with 20 μg ml⁻¹ human type IV collagen, at shear stress of 0.01 pN μm⁻². Real-time accumulation was monitored using GFP fluorescence. One representative experiment out of 12 replicates is shown. (E) The number of bound bacteria in the field of view in the accumulation experiments described in panel (D). The error bars give 95% confidence intervals of 12 replicates.

reduction of adherence, respectively, compared to untreated bacteria. To verify if the observed reduction of adherence due to Cm treatment is not a result of inhibition of protein synthesis, we used in control flow experiment Dr+ bacteria pre-incubated with kanamycin at 1000 μM as an additional control. Kanamycin-treated bacteria adhere to type IV collagen at level similar to the Dr+ bacteria not treated with antibiotics (Fig 4D and 4E).

To compare detachment kinetics with the analogous experiments on T24 cells, we reperformed the flow detachment experiments in which a low-shear (0.01 pN μm$^{-2}$) accumulation period was followed by a steady increase in shear stress from 0.58 to 9.23 pN μm$^{-2}$. Here, 97 ± 20%, 94 ± 19% and 89 ± 19% of bacteria remained stably bound on 20, 2 and 0.2 μg ml$^{-1}$ collagen coatings, respectively (Fig 4C and S7 Video, S4 Fig shows statistics for each value of shear stress employed). A comparison with the corresponding result in Figs 2 and 3 strongly indicates that the general adhesive properties of Dr+ bacteria to DAF and to type IV human collagen are the same. In addition, the observed reduction of bacterial adherence to type IV collage as a consequence of both (a) lowering the concentration of collagen and (b) lowering the number of exposed DraE receptor binding sites by Cm treatment highly supports the involvement of multivalent binding in DraE-mediated adherence.

## High shear stress induces moving of Dr+ bacteria on the surface of T24 cell layer during initial stages of adherence

To determine a general pattern of adherence of Dr+ *E. coli*, we tracked the trajectories of individual bacteria during their motion on the surface of T24 cells. Under the analyzed range of shear stress, bacteria adhered to the confluent layer of host cells to form a pattern typical for the diffusely adhering *E. coli* (DAEC) pathotype, identically as in the case of stationary adhesion (cf. Fig 1D–1F and S1 Video). In experiments using a shear stress of 0.01 and 0.06 pN μm$^{-2}$, almost all adherence events were abrupt: after the initial contact, further repositioning did not exceed 2 μm. At higher flows some bacteria initially move on the cell surface, and then either bind firmly or detach (Fig 5). The percentage of bacteria that were classified as moving after initial binding is ca. 20% and 50% at shear stresses of 0.14 and 0.58 pN μm$^{-2}$, respectively. Most bacteria move straight in the direction of flow, unperturbed by the cell line morphology. At a shear stress of 0.14 pN μm$^{-2}$, bacteria move on average over ca. 35 μm, a distance that extends to ca. 75 and 150 μm at shear stresses of 0.28 and 0.58 pN μm$^{-2}$ (Fig 5). As indicated by the presented trajectories, bacterial movement on the cell surface is complex and may include rolling, sliding and bouncing. Bacteria move at a dynamically changing velocity from <1 μm s$^{-1}$ up to 30 μm s$^{-1}$, with minor differences in velocity distribution at different values of shear stress.

In the entire range of analyzed flow rates, bacteria binding to type IV collagen stop abruptly and attach firmly to the flow cell coating before they displace by more than 3 μm. The lack of rolling on the flat surface of type IV collagen-covered flow cells suggests that rolling might result from flow perturbations due to cell layer surface morphology, assuming–as shown previously–that binding to collagen and DAF is mediated by similar mechanisms.

## Budding of T24 cells induces efficient adherence of Dr+ *E. coli* at elevated shear stress

The bladder urothelium is composed of a single layer of large (20–150 μm in diameter), highly differentiated superficial cells, covering two or three layers of small, relatively undifferentiated basal and intermediate epithelial cells. The smoothness of the superficial facet cell layer is crucial for the proper functioning of bladder. Urothelial roughness of any etiology, including infection-induced massive exfoliation, carcinogenesis and physical injuries, affects the

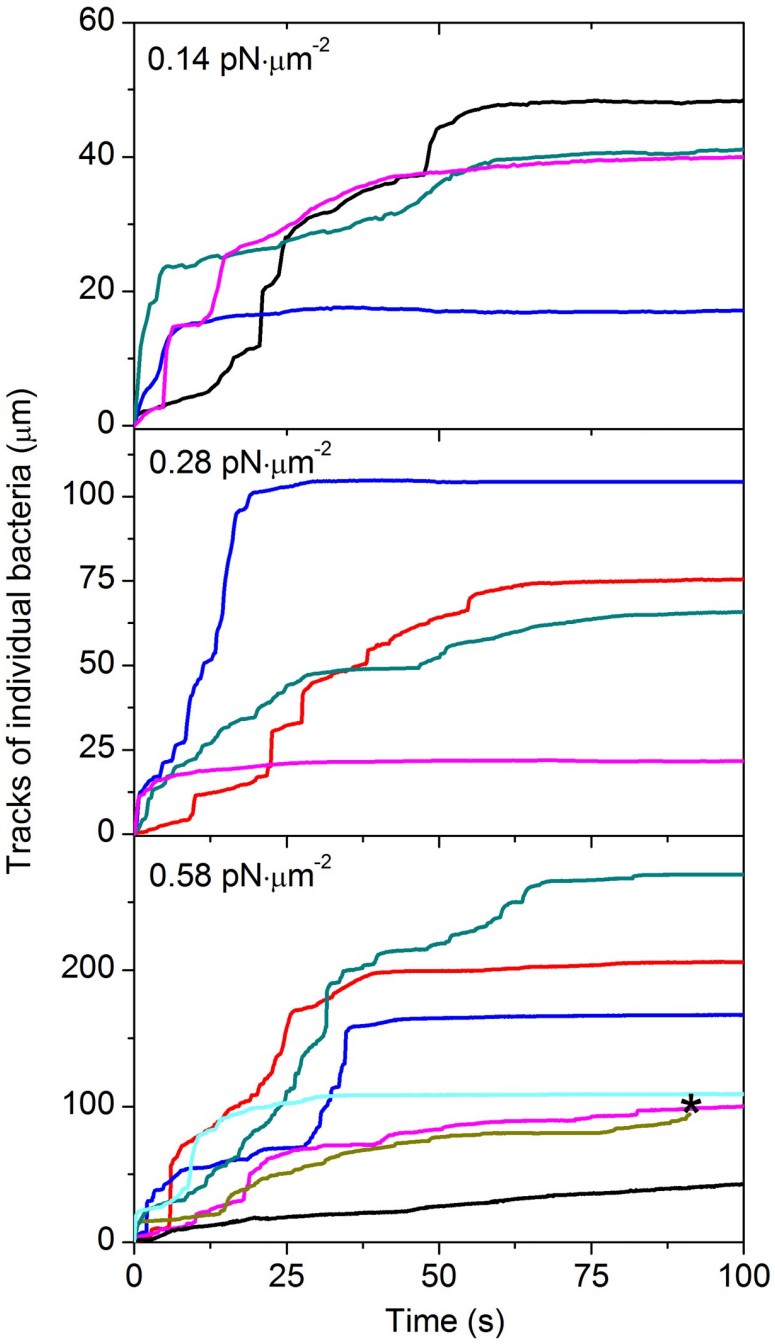

**Fig 5. Shear stress induces moving of Dr+ bacteria on the surface of the T24 cell layer.** Representative 1-dimensional trajectories of individual bacteria bound to the confluent layer of T24 cells at shear stress of 0.14, 0.28 and 0.58 pN μm$^{-2}$. An asterisk (*) indicates detachment event. Trajectories were obtained from time-lapse videos of Dr+ *E. coli* bound to T24 cells, as described in Fig 2A.

microfluidics of urinary tracts. In a mouse model, six hours after inoculation with a clinical isolate of UPEC NU14, massive exfoliation of the superficial cells was observed [38,39]. The peeling off of superficial cells results in exposition of the underlying undifferentiated urothelial cells, and even the basement membrane, so that the exfoliating urothelium has a rough structure [38,40,41]. To verify influence of "roughness" of the epithelial surface on bacteria

adherence, we analyzed the impact of flow on the binding of Dr+ *E. coli* to an overgrown layer of T24 cells lines containing 24 ± 8 single and complexes of budding cells in the field of view. In our model roughness of epithelium is generated by natural processes of cells budding and detachment and results in morphology that dimensionally may resemble naturally occurring disorders (S8 Video). The budding cells emanating from the cell layer resemble a half- sphere with a diameter of 20–30 µm, forming an obstacle that perturbs medium flow. At low shear stress (up to 0.06 pN µm$^{-2}$), bacterial attachment is not influenced by the budding cells, resulting in an adherence pattern identical to that on a confluent layer. At shear stresses ranging from 0.14 to 0.58 pN µm$^{-2}$, however, an increasing accumulation of bacteria can be observed behind the budding cells. This bacteria-enriched zone extends for over 200 µm, with a width comparable to the budding cell diameter (Fig 6A and 6D, S9 and S10 Videos). At shear stress of 0.42 pN µm$^{-2}$, the rate of adherence in this zone is 33 ± 11 mm$^{-2}$s$^{-1}$ (S5 Fig), compared to ca. 1 mm$^{-2}$s$^{-1}$ in the non-perturbed area (Fig 6C and 6D, S9 and S10 Videos). Washing of Dr+ *E. coli* for 20 minutes at a shear stress of 0.42 pN µm$^{-2}$ through flow chambers with overgrown T24 cell layer containing 24 ± 8 budding cells in the field of view resulted in accumulation of 1400 ± 415 bacteria (Fig 6D and 6F). In analogical experiments, washing of Dr+ *E. coli* over a confluent cell layer (no budding cells) resulted in 73 ± 30% reduction of bacterial accumulation compared to a T24 cell layer abundant in budding cells (Fig 6C and 6F). Washing of Dr- *E. coli* through a flow chamber with T24 cell layer containing 24 ± 8 budding cells in the field of view resulted in accumulation of <1% of the value observed for the Dr+ strain (Fig 6B and 6F), indicating that the effect depends on the presence of Dr fimbriae.

## Budding host cells force bacteria adherence by changing local fluid flow

To study the mechanism of bacterial deposition behind the budding cells, we performed numerical simulations using Fluent (ANSYS INC.) of the flow around a half-spherical obstacle with a radius of 10 µm (O$_1$), 20 µm (O$_2$) or 30 µm but centered 10 µm below the surface (O$_3$), emulating an obstacle formed by 1–3 budding cells. In reversible Stokes flow (Re<<1) and on symmetric obstacles, flow lines that start at a given distance from the surface approach the surface at the same distance after passing the obstacle. Hence, the bacterial accumulation cannot be explained by bringing flow lines closer to the cell monolayer behind the obstacle, or a decrease in flow velocity. The difference between flow and bacteria velocity, calculated using Faxen's law $\Delta u = \frac{r^2}{6}\nabla^2 u$, is also small (< 1 µm s$^{-1}$) and nearly tangent to the cell line surface when bacteria are modelled as spheres of r < 1 µm [42]. We could not rule out the possibility that the non-spherical geometry of *E. coli* cells induces rotation that brings the bacteria closer to the monolayer, but we consider it a minor effect.

Instead, our model indicates that bacteria are captured by the budding cell when flow lines concentrate near the top of the obstacle, so that the bacteria-rich fluid comes within the "capturing distance" from the cell surface. After the being captured, a fraction of bacteria remains bound to the rear side of the cell (Fig 6A and 6G, S10 Video), while others roll along the cell monolayer, creating adherence belts like those shown in Fig 6A and 6D, S10 Video. To estimate the number of bacteria that hit the budding cell per unit time in this scenario, we calculated the effective capturing cross-section of the obstacle. To this end, we considered only the flow lines that are initially outside of the assumed "capturing distance" of 1 µm from the cell surface, but enter the capturing zone upon approaching the obstacle (see Methods for details). The calculated cross-sections are marked in blue in Fig 6E, and have surfaces S$_1$ = 11 µm$^2$, S$_2$ = 19 µm$^2$, and S$_3$ = 25 µm$^2$ for obstacles O$_1$, O$_2$, and O$_3$, respectively. The flux of bacteria through a cross-section S is equal to the product of S, the density of bacteria ρ and the velocity of the fluid averaged over the cross-section. Assuming that flow velocity changes linearly with

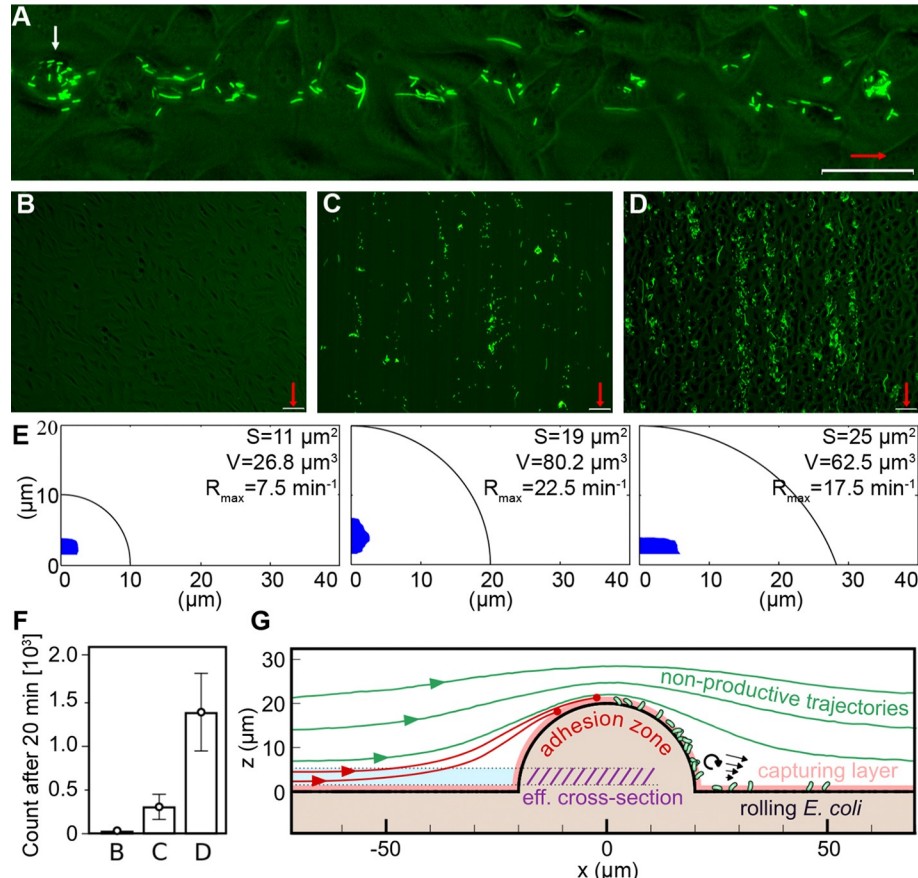

**Fig 6. Budding cells act as obstacles that induce intensive adherence of Dr+ bacteria at elevated shear stress.** (A) Formation of an increased adherence zone behind a host cell budding from cell layer (white arrow). The representative picture was taken after 8 minutes of flow, at shear stress of 0.42 pN μm$^{-2}$. Video S10 represents time-lapse videos of zone formation presented in this panel. (B) Representative picture of Dr- *E. coli* washed through the flow chamber with an overgrown layer of T24 cells with budding cells, at shear stress of 0.42 pN μm$^{-2}$. Only several bacteria were accumulated in the field of view after 20 minutes of flow. (C) Representative picture of Dr+ *E. coli* washed through the flow chamber with a confluent layer of T24 cells (no budding cells) at shear stress of 0.42 pN μm$^{-2}$. About 320 bacteria were accumulated in the field of view after 20 minutes of flow. (D) Representative picture of Dr+ *E. coli* washed through the flow chamber with an overgrown layer of T24 cells with ca. 30 budding cells, at a shear stress of 0.42 pN μm$^{-2}$. About 1800 bacteria were accumulated in the field of view after 20 minutes of flow. In panels (A-D) red arrows indicate the direction of flow, bars correspond to 50 μm. (E) Effective cross-sections (blue) calculated for the half-spherical obstacles $O_1$, $O_2$ and $O_3$ (black contour). Cross-section area *S*, volume coefficient *V* and the maximal deposition rate $R_{max}$ are shown for each simulated system. (F) The number of bound bacteria in the field of view in the accumulation experiments described in panels (B), (C) and (D). In (B) and (D), there were 24 ± 8 budding cells. The error bars give 95% confidence intervals of 12 replicates. (G) Schematic depiction of the postulated mechanism of enhanced bacterial capture by adhesive obstacles. Among flow lines that start beyond the 1 μm-thick capturing layer (shown in light red), a fraction will enter the capturing layer near the top of the obstacle (red lines) and then roll over the cell surface, while others will avoid the obstacle (green lines). The region from which the productive (red) trajectories emerge is defined as the effective cross-section (light blue). Other factors such as vorticity might additionally enhance accumulation behind the obstacle.

distance from the surface we found that the deposition rate is proportional to "volume coefficient" $V = S \times H$, where *H* is the distance of center of *S* from the boundary. This gives a maximum deposition rate $R_{max}$ (reached when all bacteria are captured) equal 7.5 min$^{-1}$, 22.5 min$^{-1}$ and 17.5 min$^{-1}$ for $O_1$, $O_2$, and $O_3$ at a shear stress of 0.42 pN μm$^{-2}$ and experimental bacteria density $\rho = 10^7$ ml$^{-1}$. The corresponding experimental adherence rates vary between protrusions, but are of the order of 8.2 ± 6 min$^{-1}$ (S5 Fig), yielding a satisfactory agreement with the

theoretical predictions. We note that while there is no experimental evidence of adherence rate increasing linearly with flow velocity, lack of this effect can be easily explained by the decreased probability of binding at high shear stress. Importantly, our numerical model is solely based on the geometry of the obstacle, managing to reproduce the observed patterns of attachment without referring to chemical or biological properties. This highlights the importance of urothelium microstructure and its disorders in shaping the microfluidic environment of urinary tracts.

## Rolling of T24 cells facilitates the spread of Dr+ *E. coli* at elevated shear stress

The host defense mechanism involves intense exfoliation of the bladder epithelium to eliminate cell-bound bacteria during urination [43,44]. Urine collected from patients with UTIs contained single or clustered exfoliated and desquamated bladder cells bound by bacteria [40,41]. However, little is known about the impact of flow on the elimination of exfoliated host cells and bacteria attached to their surface. To model this process, we investigated the effect of shear stress on the behavior of detaching host cells in a medium containing Dr+ *E. coli*. Here we assume that detached single or clustered budding T24 cells, generated by shear stress, model sufficiently well the size of naturally exfoliated and desquamated bladder cells.

At shear stress exceeding 0.58 pN $\mu m^{-2}$, the drag force detached the budding host cells, causing them to roll on the surface of a confluent monolayer. These detached epithelial cells rolled thanks to a coating of Dr+ *E. coli* that attached them to the underlying DAF-exposing monolayer. As the shear force on the detached host cell was large, the cell continued to move, shedding bacteria onto the intact monolayer. This rolling-shedding colonization proceeded indefinitely as the fluid flow supplied new bacteria to be briefly trapped on the rolling cell surface. In a repeating cascading event, briefly attached bacteria (forming few DraE-DAF contacts) were transferred onto the DAF-rich monolayer beneath, forming long stripes of adherent bacteria (Fig 7A–7C, S9 and S11 Videos). Washing of Dr+ *E. coli* for 20 minutes at a shear stress of 1.15 pN $\mu m^{-2}$ through a flow chamber with a confluent (no budding or rolling cells) T24 cell layer resulted in accumulation of 65 ± 12 bacteria (Fig 7E and 7G). At the same flow conditions, but on an overgrown cell line with budding cells, the detachment and rolling of 23 ± 8 host cells within the field of view induced accumulation of 2093 ± 386 bacteria (Fig 7F and 7G). In contrast, washing of Dr- *E. coli* through a flow chamber with a comparable number of detaching host cells resulted in accumulation of only 20 ± 5 bacteria (Fig 7D and 7G). This indicates that the estimated rate of adherence due to the intensive rolling-shedding of host cells was ca. 35-50-fold higher than on a non-perturbed confluent cell mono-layer.

When shear stress was reduced to 0.42 pN $\mu m^{-2}$ or less, the host cells stopped rolling and started to act as a budding cell-like obstacle, again inducing bacterial adherence. Conversely, an unphysiological shear stress of 2.31 or 4.62 pN $\mu m^{-2}$ resulted in full detachment of the rolling cells, along with bacteria bound to them (S3 Video). The transition between the three stages–stationary obstacle, surface rolling and full detachment–depends on the relationship between the shear and attaching forces: the former due to flow, the latter provided by the glue-like interfacial layer of adherent bacteria. By combining the effects of budding and rolling-shedding, our model explains how perturbations in the microstructure of the urothelium as well as the rolling of detached host cells affects the microfluidic environment and, consequently, greatly facilitates bacterial adherence (Fig 7H). Due to lack of adequate literature reports, though, it is difficult to estimate to what extent our *in vitro* budding rolling-shedding model of bacteria adherence adequately models natural processes occurring in the urinary tract.

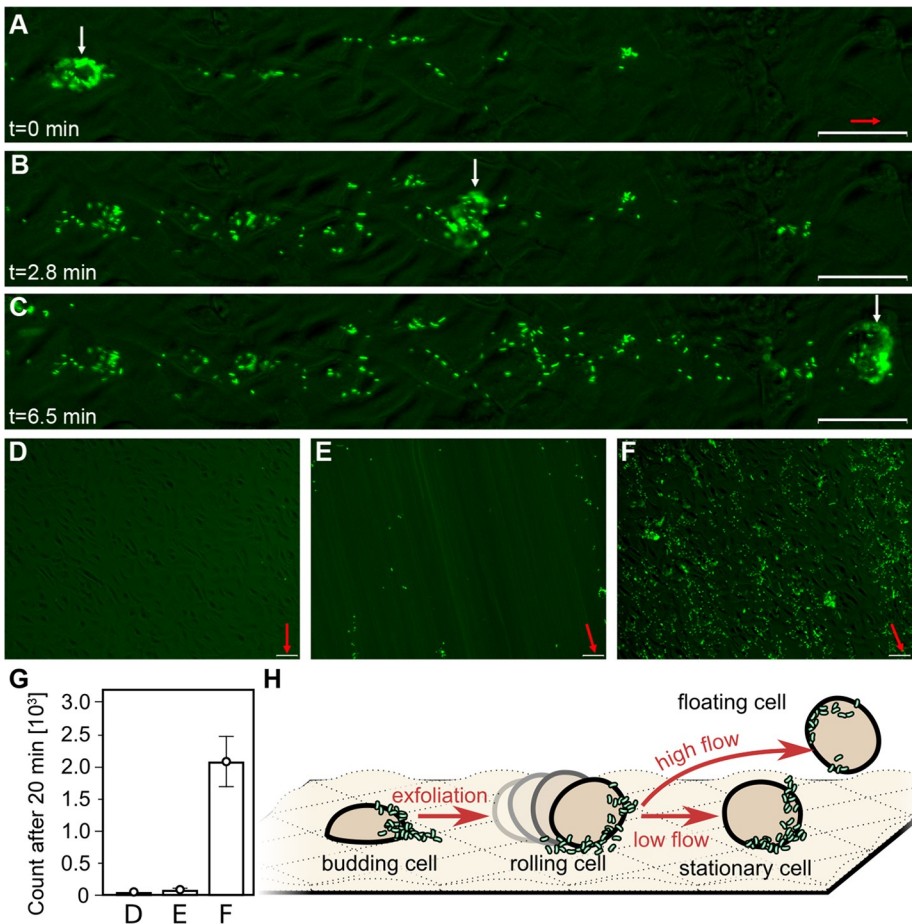

**Fig 7. Rolling host cells induce intensive adherence of Dr producing bacteria at elevated shear stress.** (A-C) Formation of an increased adherence zone due to rolling of a detached host cell at shear stress of 1.15 pN μm$^{-2}$. White arrows mark the positions of the rolling cell at a given time. Video S11 represents time-lapse videos of zone formation presented in panels (A-C). (D) Representative picture of Dr- *E. coli* washed through the flow chamber with an overgrown layer of T24 cells with budding and rolling cells at shear stress of 1.15 pN μm$^{-2}$. Only several bacteria were accumulated in the field of view after 20 minutes of flow. (E) Representative picture of Dr+ *E. coli* washed through the flow chamber with a confluent layer of T24 cells (i.e., without budding and rolling cells) at shear stress of 1.15 pN μm$^{-2}$. About 65 bacteria were accumulated in the field of view after 20 minutes of flow. (F) Representative picture of Dr+ *E. coli* washed through the flow chamber with overgrown layer of T24 cells and about 20 rolling host cells at shear stress of 1.15 pN μm$^{-2}$. About 2200 bacteria were accumulated in field of view after 20 minutes of flow. In all pictures red arrows indicate the direction of flow, bars correspond to 50 μm. (G) The number of bound bacteria in the field of view in the accumulation experiments described in panels (D), (E) and (F). In panels (D) and (F), 23 ± 8 rolling host cells moved through the field of view. The error bars give 95% confidence intervals of 12 replicates. (H) Schematic depiction of possible outcomes of cell budding/exfoliation in the flow of bacteria-rich medium. The initially detached host cell remains attached to the cell line with a layer of adherent bacteria, and rolls along the surface pushed by the flow. It captures the incoming bacteria, acting as an obstacle, and deposits them on the cell layer surface as it rolls. Eventually, high flow or low bacterial concentration might lead to full detachment, or the cell can come to a full stop in low flow conditions, accumulating more bacteria as a typical obstacle.

## Discussion

### Model of *E. coli* Dr+ adherence under shear stress

Dr fimbriae are linear non-covalent homopolymers of DraE adhesin subunits assembled via the chaperone-usher pathway [9,12]. Our earlier works highlighted the exceptional resistance of Dr fimbriae to physical and chemical denaturation. This stability stems from the structural

properties of a single DraE protein that melts at ca. 87˚C, with a free energy of unfolding of 83 kJ mol$^{-1}$ and a unfolding half-life of $10^8$ years at 25˚C [45,46]. We concluded that this high stability, typical for adhesive structures of the chaperone-usher type, is crucial for the maintenance of integrity and functionality of Dr fimbriae [47]. However, colonization of the urinary tract by Dr+ *E. coli* is equally dependent on the formation of productive bonds between Dr fimbriae and host receptors. Indeed, the extreme stability of the DraE polymer would be of little use if the adhesin-receptor complex failed to withstand the shear stress caused by urine flow.

Dr fimbriae recognize multiple receptors including DAF, members of the CEACAM family and type IV collagen, even though only DAF is consistently recognized by all 13 members of the Dr adhesin family [48,49]. Consequently, DAF is considered to be the main receptor responsible for the initial adherence of Dr+ bacteria in the urinary tract [18]. Other receptors, recognized only by a subset of the Dr family adhesins, presumably modulate the host- and strain-specific process of pathogenesis [13,18]. To investigate the effect of shear stress on the initial stage of adherence of Dr+ bacteria, we used T24 bladder transitional cells as a model host cell line that only exposes a single DraE receptor, the DAF protein. Using this minimal system, we showed that Dr+ *E. coli* most efficiently adhere to T24 cells under static conditions, and that any increase in shear stress reduced number of bound bacteria. In the physiologically relevant range of shear stresses (from 0.017 pN μm$^{-2}$ in the proximal renal tubule to 0.3–0.5 pN μm$^{-2}$ in the urethra) we observed a reduction in the accumulation rate by 70%. Still, at shear stresses up to 0.58 pN μm$^{-2}$, about 80% of the initial binding events resulted in stable adhesion, with the remaining 20% detaching usually within few second from the initial binding. This indicates that the observed 70% reduction in the accumulation rate mainly results from the inhibition of initial attachment due to medium flow. The drag force acting on the adherent bacteria can be approximated using the Goldman's modification to the Stokes' formula, relevant for a sphere near the surface: $F_D = 1.7 \cdot 6\pi r^2 \tau$, where $r$ is the sphere radius (approximately 1 μm for *E. coli*), and $\tau$ is the shear stress at the surface [50]. As the shear stress increases from 0.017 pN μm$^{-2}$ in the proximal renal tubule to 0.5 pN μm$^{-2}$ in the urethra, the drag force to be withstood by the adhesin-receptor complex changes from 0.5 pN to 16 pN. If we assume that the DraE-DAF binding affinity is independent of shear stress, as in the slip-bond mechanism, stable adhesion at faster flow will require the formation of more receptor-ligand complexes to balance the drag force.

We note that the initial stage of adherence necessarily engages only a limited number of DraE subunits from the outermost layer of the adhesive Dr envelope. The number of these apical subunits defines the maximal strength of the interaction during initial stages of adherence. Thus, as we increased the shear stress, the number of productive initial binding events decreased until no adherence was observed at 2.31 pN μm$^{-2}$. To confirm that shear stress inhibits the adherence of Dr+ *E. coli* by reducing the success rate of initial binding events, we blocked the interaction between DraE and DAF using chloramphenicol (Cm). Bacteria pre-incubated with 100 and 200 μM Cm yielded 30% and 85% less initial attachment events at the lowest shear stress of 0.01 pN μm$^{-2}$, respectively. A similar behavior–a large reduction in the number of initial binding events–was observed with decreasing concentrations of type IV human collagen, emulating a decreasing number of exposed host receptors. It is worth noting that virtually all bacteria that survived the initial attachment period subsequently remained stably bound in both types of experiments, indicating that bacteria only require few mere seconds to establish an almost permanent mode of adhesion. We confirmed this with the detachment experiments in which pre-bound Dr+ bacteria were treated with Cm under increasing drag force. Even under the harshest conditions of 2 mM Cm and shear stress of 9.23 pN μm$^{-2}$, only 30% of bacteria eventually detached. This means that the transition from initial to stable

adhesion involves the creation of multiple interactions that confer high binding stability. We propose two non-exclusive mechanisms to explain this effect. In scenario (i), only few outermost DraE subunits of the Dr envelope are engaged in the initial binding, with more DraE--DAF contacts formed subsequently by both the remaining surface located DraE subunits as well as ones located deeper in the adhesive Dr envelope; the latter effect can be facilitated by the drag force itself that mechanically deforms the adhesive capsule, exposing the deeper located DraE subunits (Fig 8). In scenario (ii), the adhesion of Dr+ *E. coli* to the host cell induces clustering of DAF around the bound bacteria [14,51,52].

We came to the conclusion that Dr-fimbiated *E. coli* adhere to the host cells by bonds that become shorter lived when subjected to a tensile force. This results in the high sensitivity of the initial step of binding to both shear stress and the microarchitecture of the epithelium. The observed stable, almost permanent adhesion of bacteria at a later stage of binding is, in contrast, caused by multivalent interaction of Dr fimbriae with cell-exposed receptors. This effect is exclusively dependent on the polyadhesive properties of DraE polymers with receptor binding sites distributed uniformly throughout the capsule, and is consistent with identical adhesion patterns observed for DAF and type IV collagen, two receptors with a completely different structure. In conclusion, the adherence of Dr+ *E. coli* to DAF and type IV collagen is based on the slip-bond model of interactions that is significantly influenced by the polyadhesive nature of Dr fimbriae. This model of adhesion contrasts with that described for *E. coli* strains producing monoadhesive type 1 pili that use catch-bonds, strengthening under shear stress due to structural changes within the FimH-mannose complex. Raising the shear stress above a certain threshold actually increased the accumulation of type 1-piliated bacteria, and reducing the shear stress below the threshold resulted in their detachment [28].

## Experimental model of adherence of Dr+ *E. coli* under shear stress in the context of UPEC pathogenesis

To investigate bacterial colonization during mimicking natural host environment, we set up an *in vitro* micromodel that reproduces several key physiological properties of the human urinary tract (UT). Firstly, we emulated UT segment-specific shear stresses and flow velocities from the renal tubules through ureters and bladder to the urethra. We used transitional epithelial cells, lining the lumen of human ureters, bladder and part of the urethra, to mimic natural epithelial conditions including expression levels of apical DAF. In turn, the use of type IV collagen–a component of the basement membrane located beneath the uroepithelial layer–represents a deep tissue receptor that becomes accessible for attachment following epithelial desquamation due to shear stress, injury during urologic procedures or interstitial invasion of UPEC. Finally, we selected *E. coli* bearing Dr fimbriae due to their recognized role in chronic, recurrent and often subclinical infection [8].

**DAF expression levels in UTI pathogenesis.** In the urinary tract, Dr fimbriae only interact with human versions of DAF receptor [53]. Therefore, there is no suitable animal model to study the pathogenesis of Dr+ UPEC strains. In this setting, the most important reference data come from clinical description of UTIs caused by DAEC strains expressing Dr or Afa-III adhesins. Dr/Afa-III producing UPEC are most often associated with chronic pyelonephritis in the third trimester of pregnancy in women with gestational complications [54,55]. It is suggested that the pathogenesis mediated by Dr+ UPEC depends on the availability and abundance of DAF. Predisposition to Dr+ UPEC-related UTIs in pregnant women likely results from a physiological increase in DAF expression level during pregnancy, a means of protection of the semi-allogenic fetus against complement attack [56–58]. Additionally, factors that modulate DAF expression–endogenous nitric oxide (NO) synthesis or the phosphoinositide 3-kinase/

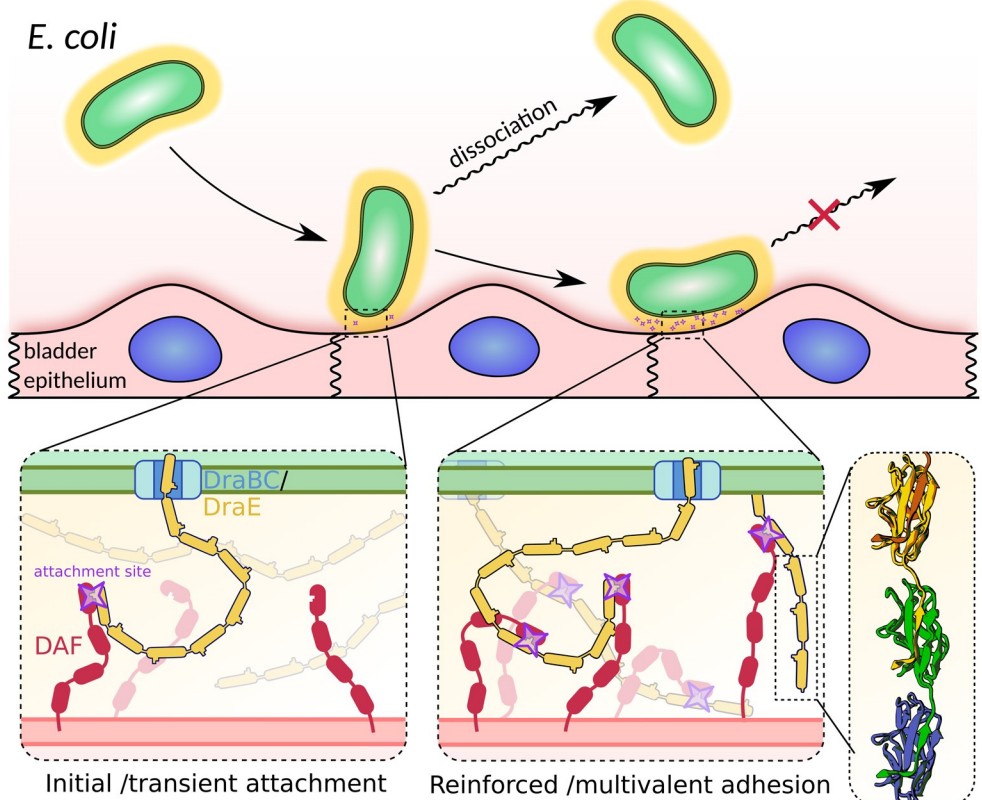

**Fig 8. Model of adherence of Dr producing *E. coli* cells to T24 bladder cell lines at shear stress generated by fluid flow.** Formation of DraE-DAF contacts upon initial contact between the bacterial and human cells. Transiently bound bacteria are anchored by few such interactions and are hence prone to dissociation in high shear stress conditions. Within seconds, the formation of further adhesin-receptor complexes reinforces the adherence, leaving the bacteria almost permanently attached to the cell line.

protein kinase B (PI3K/Akt) pathway–were shown to modulate the adhesion and invasion of Dr-piliated bacteria [59,60]. These data are consistent with our model of Dr+ bacterial adherence to the host cells under flow conditions, highlighting the critical role of DAF concentration in the initial stage of adherence. In the model, bacterial adherence is maximal under low shear stress, conditions typical for e.g. renal tubules.

**Mode of pathogenesis vs adherence system.** Type 1-piliated UPEC isolates cause acute bladder infections with intense exfoliation of the superficial umbrella cells. This allows these bacteria to invade deeper layers of bladder epithelium, form a biofilm-like intracellular bacterial communities and finally escape from the harsh environment of bladder surface [61,39]. For example, the fecal F18 *E. coli* isolates that induce acute cystitis possess a FimH variant that binds to monomannose through the catch-bond mechanism [28]. This permits the bacteria to detach from the exfoliated host cells and re-bind to the newly exposed layers of the bladder epithelium in order to reduce the number of bacteria washed off with dead host cells.

Unlike type 1-piliated bacteria, Dr+ isolates would not benefit from intensive exfoliation, until invasion to deep interstitial collagen receptor occurs, as these strains only adhere to and invade fully differentiated superficial facet cells lining the luminal walls of the bladder, ureter and renal pelvis; the underlying immature cells that become exposed during exfoliation do not produce DAF and CEACAM [62]. Accordingly, the interaction of Dr/Afa-producing bacteria with CEACAM proteins highly suppresses epithelial exfoliation [63]. In our flow experiments,

we have shown that Dr+ bacteria bind almost permanently to DAF-exposing host cells *via* the slip-bond mechanism, i.e., without a minimal shear stress threshold, so that once bound, the bacteria do not detach from the host cells during periods of halted flow. Meanwhile, the multivalent adherence mediated by DraE-DAF interactions is sufficiently strong to allow the bacteria to persist attached to the UT luminal walls under conditions of urine flow.

**Effect of disruption of the epithelial microarchitecture on bacterial colonization.** In contrast to *in vitro* conditions, the surface architecture of UT luminal walls undergoes frequent changes due to peristaltic waves of ureters or contraction and dilatation of the bladder. Likewise, the microarchitecture of distensible luminal epithelium changes constantly. In addition, the cell microarchitecture is altered by pathophysiology such as apoptotic blebs formation, desquamation and regeneration. We therefore incorporated the role of budding cells of the apical epithelial surface, the budding cells, into our model of bacterial adherence. We observed that budding host cells facilitate the adherence of bacteria at flow conditions typically found in the UT, whereas the binding to a smooth cell layer is severely limited. At shear stresses ranging from 0.14 to 0.58 pN $\mu m^{-2}$, we observed an increasing accumulation of bacteria behind the budding transitional cells. The numerical model of flow disturbance by half-spherical obstacles showed that the observed preferential adherence results from the concentration of flow lines atop the obstacle, and is a general feature solely dependent on the geometry of the system. This suggests that practically any form of epithelial roughness, including carcinogenesis, genetic defects or physical injuries, may facilitate bacterial adherence and contribute to the pathogenesis of UTIs.

**Effect of exfoliated epithelial cells.** Exfoliated cells in the UT either come from physiological urothelium turnover or enhanced exfoliation of different etiology, including injury, carcinogenesis or UTI. To model an exfoliating urothelium, we used overgrown T24 cell line in which individual cells start to detach from the monolayer. We showed that Dr+ bacteria do not have the ability to detach from dead free floating host cells, in contrast to type 1-piliated UPEC. As a consequence, the detached T24 cells observed in flow experiments are covered in a mesh of Dr+ *E. coli*. These bacteria will also provide adherence to the monolayer, either anchoring the detached host cells at low flow or causing them to roll on the monolayer surface when flow velocity is high. Unexpectedly, both phenomena will induce intense bacterial adherence to the epithelial layer at shear stress that would otherwise reduce the attachment rate. Indeed, one would intuitively expect the urine flow to efficiently eliminate the infected exfoliated cells, thereby protecting the UT from chronic and recurrent UTIs. In stark contrast, the *in vitro* rolling-shedding-refilling (RSR) model shows how the wash off of detached colonized cells will instead provoke a massive spread of *E. coli* adherence and facilitate colonization. Interestingly, a related rolling behavior was observed for flow-induced movement of erythrocytes along a biofilm formed by type 1-piliated *E. coli* [23]. The fact that catch-bond interactions between FimH and mannose can also sustain surface rolling suggests that our micromodel can be applicable more broadly to explain aspects of adhesion-dependent UTIs. The described here micromodel implicates the RSR colonization mode in the mechanism of recurrent or protracted forms of UTI, as well as in common therapeutic failure. Finally, since animal models have significant limitations our micro-model may be instrumental in investigation of novel therapeutic modalities including antibiotics and/or attachment blockers prior to clinical trials.

## Materials and methods

### Bacterial strains and plasmids

*E. coli* IH11128 is a human pyelonephritis-associated prototype UPEC strain carrying the *dra* gene cluster [64]. *E. coli* AAEC191 is a derivative of the laboratory K12 strain. The strain has a

deletion of the entire *fim* operon (Δ*fim*) made by allelic exchange, preventing the expression of type 1 pili [65]. It was used for the production of Dra proteins (encoded by a pCC90 plasmid) required for Dr fimbriae biogenesis.

The pCC90 plasmid contains the *dra* gene cluster with its promoter and regulatory genes upstream of *draB* gene deleted [17]. Bacterial strains encoding this plasmid produce native Dr fimbriae on the cell surface. The pCC90D54stop plasmid (DraE-negative) is a derivative of pCC90 with a mutated *draE* gene (a codon for D54 in DraE changed to a stop codon) used as a negative control of Dr fimbriae production [17].

pSFOXB20-daGFP (pGFP) is a commercial vector encoding GFP (Green Fluorescent Protein) controlled by the OXB20 constitutive promoter. The plasmid bears an origin of replication derived from pBR322 and a kanamycin resistance gene (Oxford Genetics). It was used to visualize the bacterial cells using fluorescence microscopy.

In this study, we used *E. coli* AAEC191/Dr/GFP (labeled Dr+) and AAEC191/Dr-stop/GFP (labeled Dr-) strains bearing pCC90/pSFOXB20-daGFP and pCC90D54stop/pSFOXB20-daGFP plasmids, respectively. For each experiment, bacterial cells were grown in Luria broth (LB) supplemented with the appropriate antibiotics (100 μg ampicillin ml$^{-1}$ and 20 μg kanamycin ml$^{-1}$; Sigma-Aldrich).

### Cell lines

The T24 (ATCC HTB4) human urinary bladder cell line was purchased from ATCC-LGS (USA). The cells were cultured in McCoy's 5a medium supplemented with 10% (vol/vol) fetal bovine serum (FBS) (PAN-Biotech) and penicillin/streptomycin solution (PAN-Biotech) in a 5% $CO_2$ atmosphere at 37˚C. The cells were passaged using 0.25% (vol/vol) trypsin-EDTA (PAN-Biotech).

### Antisera

**Detection of the Dr fimbriae in the laboratory AAEC191/Dr/GFP and clinical IH11128 UPEC strains.**   Rabbit anti-Dr serum directed against purified wild type Dr fimbriae were purchased from Immunolab (Gdynia, Poland). Goat anti-rabbit IgG (whole molecule) labeled with horseradish peroxidase and TRITC mouse anti-rabbit secondary antibodies were purchased from Sigma-Aldrich and used in the Western blotting and immunofluorescence assays, respectively.

**Immunofluorescence of DAF (CD55) on the surface of T24 cell line.**   Immunofluorescence-dedicated polyclonal mouse anti-DAF antibodies raised against full-length human DAF protein (catalog number: SAB1405695, Sigma-Aldrich) and donkey anti-mouse IgG (H+L) cross-adsorbed secondary antibodies, DyLight 488 (catalog number: SA5-10166, Thermo Fisher Scientific) were used.

**Flow cytometry detection of DAF and CEACAM family proteins on the T24 cells.** FITC mouse anti-human CD66 clone B1.1/CD66 antibodies recognizing: CD66a (CEACAM1), CD66c (CEACAM6), CD66d (CEACAM3) and CD66e (CEACAM5) antigens (the molecules of carcinoembryonic antigen, CEA family) (catalog number: 551479, BD Biosciences), FITC mouse anti-human CD55 clone IA10 antibodies specifically binding to complement decay-accelerating factor, DAF (catalog number: 561900, BD Biosciences) and FITC mouse IgG2a, κ isotype control clone G155-178 antibodies (screened for low background binding on human tissues) (catalog number: 555573, BD Biosciences) were used.

### Western blotting and immunofluorescence staining

To determine the amount of Dr adhesin on the surface of our laboratory strains: *E. coli* AAEC191/Dr/GFP, AAEC191/Dr-stop/GFP and the clinical IH11128 UPEC strain, Dr

fimbrial fractions were isolated from cells cultivated overnight in LB medium, as described earlier [66]. The same volumes (15 μl) of prepared fimbrial samples were mixed with Laemmli sample buffer (5 μl), denatured at 100°C for 60 min and ran in 15% (w/v) bis-acrylamide gels containing SDS. To visualize bands corresponding to DraE, Western blotting was performed using primary rabbit anti-Dr and secondary goat anti-rabbit antibodies labeled with horseradish peroxidase. The relative concentration of DraE in the fimbrial fractions was determined by densitometry analysis using a VersaDoc system with Quantity One software (both from Bio-Rad, Hercules, CA). The reference IH11128 UPEC strain was arbitrarily set as 100%. The experiment was repeated three times. The optical density was determined densitometrically using the average of the analyzed bands from the three measurements per each experiment.

For immunofluorescence labeling, T24 culture cells grown in 35 mm dishes were fixed in 3% paraformaldehyde (Sigma-Aldrich) in DPBS for 10 min at room temperature, washed six times with DPBS/0.01% Tween 20 (Sigma-Aldrich), then blocked for 1h in 5% BSA in DPBS and washed with DPBS/1% BSA. To label DAF antigens, the primary mouse anti-DAF and the secondary DyLight 488 labeled anti-mouse antibodies were diluted (1:50 and 1:100, respectively) in DPBS supplemented with 1% BSA and 0.01% Tween 20 and incubated with the cells for 1h at 37°C. After incubation with the primary and secondary antibodies, the cells were rinsed four times with DPBS/1% BSA/0.01% Tween 20. Alternatively, to immunolabel host cell bound Dr+ bacteria, the primary rabbit anti-Dr and secondary TRITC labeled anti-rabbit antibodies were used, diluted 1:50 and 1:100, respectively.

## Flow cytometry

T24 human urinary bladder cancer cells were harvested from the culture flask by enzymatic release using 0.25% trypsin-EDTA and re-suspended in McCoy's 5a medium with 10% FBS by gently pipetting. Then, the cells were centrifuged at 140 g for 5 min, the supernatant was discarded and the pellet was re-suspended in room temperature PBS supplemented with 0.5% FBS (to remove any cell debris). The wash off step was repeated again. Finally the cells were suspended in the appropriate amount of PBS/0.5% FBS, counted using automated cell counter (ZB1 Coulter Counter) and diluted in the same buffer to a concentration of $1 \times 10^6$ cells ml$^{-1}$. Then, the cells were incubated with staining solution containing FITC conjugates of specific antibody (anti-human CD66 and anti-human CD55) (80 μg ml$^{-1}$) or with an isotype control (IgG2a) murine mAb for 30 min on ice. After 30-min staining on ice, the cells were washed two times with PBS/0.5% BSA. Flow cytometry was performed using BD Accuri C6 flow cytometer (Becton Dickinson). An argon ion-laser was used for excitation at 488 nm and a 520 nm long-pass filter emission was used for detection of fluorescence emitted by FITC. Flow cytometric data were analyzed using BD Accurin C6 software.

## Bacterial adherence to cell culture under static condition

T24 cells were seeded in 35 mm polystyrene dishes (Corning) and grown for 24 h at 5% $CO_2$ and 37°C in McCoy's 5a medium (PAN Biotech) modified with 10% FBS (PAN Biotech) and antibiotic solution to a confluent monolayer. To obtain T24 cells with blocked DAF receptors, the cell culture was further fixed in 3% paraformaldehyde and incubated with anti-DAF antibodies (dilution 1:10) identically as described for DAF immunofluorescence. Selected dishes with cell culture were rinsed twice with fresh medium and used in the adherence assay. Prior to the adherence experiment under static conditions, the overnight bacterial cultures were centrifuged at 2000 g, suspended in McCoy's 5a medium with 0.5% BSA (catalog number: A3294-100G, Sigma-Aldrich) saturated with 5% $CO_2$ and adjusted to $OD_{600}$ of 0.1 ($10^7$ bacteria CFU ml$^{-1}$). A confluent layer of T24 cell or T24 cells with blocked DAF receptors were incubated

with 2 ml of appropriate *E. coli* strain in static condition for 30 minutes at 37°C. Unbound bacteria were washed off using McCoy's 5a/0.5% BSA medium, while bound ones were visualized via endogenous GFP fluorescence or immunofluorescence with primary rabbit anti-Dr and secondary TRITC labeled mouse anti-rabbit Ab.

## Cell line accumulation and detachment assays under flow conditions

T24 cells and bacteria were cultivated identically as in the stationary adherence assay, but growth time was extended to obtain overgrown T24 cell layers with budding cells. The selected dish containing the cell culture was rinsed twice with fresh medium and used to assemble a flow chamber with a gasket 1 cm long, 2.5 mm wide and 0.25 mm thick (GlycoTech, USA). Flow chamber adherence experiments were performed at a temperature of 25°C, at which the bacterial suspension had a viscosity of 0.0093 P.

**Accumulation of bacteria on T24 cell layer under shear stress.** Bacterial suspension was passed through the flow chamber to generate a shear stress of 0.01, 0.06, 0.14, 0.28, 0.42, 0.58 and 1.15 N $\mu m^{-2}$ for 20 min with a syringe pump (Harvard Apparatus). Then, to remove the unbound (non-adherent) bacteria, the pump was switched to culture medium for 2 min. To examine the sensitivity of bacterial adhesion to chloramphenicol (Cm), bacteria were pre-incubated in the medium supplemented with 50, 100 or 200 μM Cm (Sigma-Aldrich) for 20 min. Then the attachment to bladder cell line was analyzed under shear stress of 0.01 pN $\mu m^{-2}$ for 20 min, as described above. The experiments were performed at least four times in triplicate.

**Accumulation of bacteria on T24 cell layer with budding cells under shear stress.** The dishes with a mounted flow chamber were positioned on an inverted microscope to have from 15 to 30 ($24 \pm 8$) single budding cells or their complexes in the field of view. To investigate the formation of adherence zones behind stationary budding cells, the bacterial suspension was passed through the flow chamber to generate a shear stress of 0.42 pN $\mu m^{-2}$ (does not induces budding cells detachment) for 20 minutes. To study the influence of detached rolling host cells on bacterial adherence, the bacterial suspension was washed at a shear stress of 1.15 pN $\mu m^{-2}$ (induces budding cells detachment) for 20 minutes. Only experiments in which the number of rolling host cells ranged from 10 to 30 within 20 minutes were considered for statistical analysis. For both types of experiments, unbound bacteria were removed by washing the chamber with culture medium for 2 min. Two types of control experiments were used. To check for the influence of stationary budding or detached rolling cells on the induction of non-specific adherence, the suspension of Dr- (non-fimbriated) bacteria was passed through a flow chamber with budding cells at a shear stress of 0.42 and 1.15 pN $\mu m^{-2}$, respectively. In the second control, a suspension of Dr+ bacteria was passed through a flow chamber with a confluent layer of T24 cells with no budding cells at a shear stress of 0.42 or 1.15 pN $\mu m^{-2}$. 12 recorded experiments of each type from at last 3 different experiments were used for statistical analysis.

**Detachment of adherent bacteria from T24 cell layer under increasing flow gradient.** Bacteria suspended in the medium as described above were washed through the flow chamber with T24 cells at shear stress of 0.01 pN $\mu m^{-2}$ for 20 min. The flow was then switched to bacteria-free medium to wash unbound bacteria (shear stress of 0.01 pN $\mu m^{-2}$ for 5 min). As a result, from 1100 to 1600 bacteria were bound to T24 cells in the field of view. After the accumulation stage, bound bacteria were subjected to a stepwise increasing flow corresponding to shear stress of 0.06, 0.28, 0.58, 1.15, 2.31, 4.62, 9.23 and 18.46 pN $\mu m^{-2}$ for a total time of 28 min (from 0.06 to 4.62 pN $\mu m^{-2}$ in steps lasting 264 s each, and 9.23 and 18.46 pN $\mu m^{-2}$ lasting 65 and 30 seconds, respectively). In an alternative experiment, consecutive steps of flow were separated by pauses that lasted 2 minutes each. The experiment was further modified to examine the influence of Cm on bacterial detachment under different values of shear stress. Bacteria

were accumulated on the T24 cells in Cm-free medium as described above. After this stage, adherent bacteria were subjected to flow of medium supplemented with 300, 600 or 2000 μM Cm at increasing shear stress from 0.06 to 9.23 pN μm$^{-2}$ in eight steps lasting 180 seconds each. Both continuous- and intermittent-flow detachment experiments were performed at least four times in triplicate.

## Human type IV collagen accumulation and detachment assays under flow conditions

The 35 mm dishes were coated overnight with type IV collagen (from human placenta; Sigma-Aldrich) at concentrations of 20, 2 and 0.2 μg ml$^{-1}$ at 4°C, and blocked prior to use with DPBS (PAN Biotech) supplemented with 0.5% BSA at 37°C for 1.5 h. The bacterial cells were prepared as described above, suspended in DPBS/0.5% BSA and adjusted to $OD_{600} = 0.1$. After blocking, the dish was vacuum-mounted to the chamber and washed with DPBS/0.5% BSA under flow conditions of 0.14 pN μm$^{-2}$ for 3 min. In the accumulation flow experiments, the bacterial suspension was passed through the flow chamber under shear stresses of 0.01, 0.06, 0.14, 0.28, 0.58 and 1.15 pN μm$^{-2}$ for 16 minutes. For collagen concentration of 0.2 μg ml$^{-1}$, bacterial adherence was observed at shear stress up to 0.14 pN μm$^{-2}$. To examine the sensitivity of Dr+ *E. coli* binding to type IV collagen to chloramphenicol (Cm), bacteria were pre-incubated in the medium supplemented with 50, 100 or 200 μM Cm for 20 min. Then the attachment to dishes coated with 20 μg ml$^{-1}$ type IV collagen was analyzed under a shear stress of 0.01 pN μm$^{-2}$ for 16 min. Alternatively, to check for possible impact of protein synthesis inhibition by Cm on adherence to type IV collagen, Dr+ *E. coli* were pre-incubated with kanamycin (Kn) at 1000 μM Cm for 20 min. Kn-treated bacteria were tested on dishes coated with 20 μg ml$^{-1}$ type IV collagen under a shear stress of 0.01 pN μm$^{-2}$ for 16 min. In the detachment experiments, bacterial suspensions were washed through the flow chamber with an appropriate collagen concentration at shear stress of 0.01 pN μm$^{-2}$ for 10 minutes. Eventually, about 800, 300 and 100 bacteria bound in the field of view to collagen at concentrations of 20, 2 and 0.2 μg ml$^{-1}$, respectively. Bound bacteria were subjected to washing off under stepwise increasing flow corresponding to shear stress of 0.58, 1.15, 2.31, 4.62, 9.23 pN μm$^{-2}$ for a total time of 16 minutes. Each type of experiments was performed at least four times in triplicate.

## Calculations of flow around the obstacle and bacteria adherence rate

The experimental flow chamber was $w = 2.5$ mm wide and $h = 0.25$ mm high, with a cross section of $a = w \times h = 0.625$ mm$^2$. The bacterial suspension was passed through the chamber at volumetric flow rates: $Q = 0.02, 0.12, 0.25, 0.5, 0.75, 1$ and $2$ ml min$^{-1}$. For such a setup, the flow far from walls is parabolic:

$$u = \frac{3Q}{2a}\left(1 - \frac{z^2}{h^2}\right),$$

where $3Q/2a$ is the maximum velocity (equal to 39 mm s$^{-1}$ for flow rate 1 ml min$^{-1}$), and $z$ the distance from the symmetry plane. At the boundary, this flow generates shear $sh$:

$$sh = \frac{6Q}{ah} = \frac{640}{s} \times Q\left[\frac{ml}{min}\right],$$

where $Q$ is in units ml min$^{-1}$, and resulting shear in s$^{-1}$. For the measured dynamic viscosity of

the fluid $\mu = 9.3 \times 10^{-4}$ Pa·s, the corresponding shear stress is

$$\tau = \frac{6Q\mu}{ah} = 0.58 \frac{\text{pN}}{\mu\text{m}^2} \times Q \left[ \frac{\text{ml}}{\text{min}} \right].$$

The resulting shear stresses were: 0.01, 0.06, 0.14, 0.28, 0.42, 0.58 and 1.15 pN $\mu\text{m}^{-2}$ (with 0.58 pN $\mu\text{m}^{-2}$ corresponding to a volumetric flow rate of 1 ml min$^{-1}$).

The budding cells rise up to about 20 μm above the surface of the monolayer. The Reynolds number calculated based on the budding cell height (20 μm) satisfies Re < 1, and calculated based on height of the chamber $h$ satisfies Re <10, even for the highest volumetric flow rate of 2 ml min$^{-1}$, which implies that flow around the budding cell remains laminar.

As flow velocity changes approximately linearly with the distance from the boundary, the average velocity through the cross-section is $sh \times H$, where $H$ is the distance of center of cross-section $S$ from the boundary. Thus, the maximum deposition rate $R_{\max}$ is

$$R_{\max} = \rho \times sh \times H \times S,$$

where $H \times S$ has units of volume and characterizes a given obstacle, while $\rho \times sh$ characterizes bacteria flow. The calculated "volume coefficients" $V = H \times S$ are 26.8 $\mu\text{m}^3$, 80.2 $\mu\text{m}^3$ and 62.5 $\mu\text{m}^3$ for obstacles $O_1$, $O_2$, and $O_3$, respectively. The experimental bacteria density was estimated as $\rho = 10^7$ ml$^{-1}$.

### Recording and analysis of the bacterial adherence in time-lapse videos

The adherence of bacteria emitting green fluorescence was recorded with the Olympus IX73 inverted fluorescence microscope equipped with the UPlanFL N 10x/0.30 objective and a Hamamatsu Orcaflesh 2.8 CMOS digital camera. The field of view was 696.2 x 522.2 μm, with a resolution of 0.363 μm per pixel. Adherent bacteria were recorded in time-lapse digital videos using the Olympus cellSens Dimension 1.18 software. The videos were recorded using a shutter speed at which floating bacteria were blurred out, whereas surface-bound bacteria were detected as lighting spots. The shutter speed was adjusted to match the flow velocity, and ranged from 500 ms for shear stress of 0.01 pN $\mu\text{m}^{-2}$ to 20 ms for shear stress above 1.15 pN $\mu\text{m}^{-2}$. To count the bound bacteria in each recorded frame, the intensity threshold was adjusted manually so that fluorescent bacteria contrasted with the dark background. Bacterial counts and trajectories were determined using the 'Count and Measure' package in Olympus cellSens and the spots tracking command in Imaris x64 Bitplane AG software, respectively. The time-lapse videos were converted to the mp4 format using Kdenlive.

### Statistical analysis

The statistical analysis of the attachment/accumulation counts was performed according to ref. [28]. If the mean number of attached bacteria $n$ in a series of $k$ counting experiments was less than 100, the Poisson statistics was applied, with the confidence level of 95%, i.e. the experimental error was estimated as twice the standard deviation. In cases where $n > 100$, the standard deviation was estimated as the square root of $n$, and the error was again estimated to be twice the SD. Errors of the ratios were calculated according to the rules of error propagation.

### Supporting information

**S1 Fig. Recombinant Dr+ and clinical uropathogenic IH11128 *E. coli* strains produce Dr fimbriae at a similar level.** Representative Western blotting analysis of fimbrial fractions isolated from the: IH11128 UPEC clinical strain, laboratory Dr+ AAEC191/Dr/GFP strain and

laboratory Dr- AAEC191/Dr-stop/GFP strain. Prior to isolation of fimbrial fractions, the overnight bacteria cultures were centrifuged and resuspended in a PBS to $OD_{600}$ of 1.0. Western blotting was performed using primary rabbit anti-Dr and secondary goat anti-rabbit antibodies labeled with horseradish peroxidase. The relative concentration of DraE in the fimbrial fraction isolated from Dr+ and Dr- bacteria was determined by densitometry analysis, with the IH11128 UPEC strain used as a reference (100%). The experiment was repeated three times. The laboratory Dr+ strain produces Dr fimbriae at a level of 105 ± 12% relative to IH11128. For the Dr- strain no signal corresponding to Dr fimbriae was recorded.
(TIF)

**S2 Fig. Dr+ *E. coli* pre-bound to T24 cells are resistant to detachment by flow at high shear stress.** (A) and (B): the flow detachment experiment without and with 2-minute pauses between subsequent flow velocity steps, respectively; see Fig 2D caption for experimental details. The figure shows the fraction of bacteria that remain bound at the end of each 264-second flow step, determined as the difference between bacteria count at the start and end of each flow step. The error bars give 95% confidence intervals of 12 replicates.
(TIF)

**S3 Fig. Dr+ *E. coli* pre-bound to T24 cells are resistant to detachment by flow of medium supplemented with Cm.** (A), (B) and (C): the flow detachment experiment using flow medium supplemented with 300, 600 and 2000 μM Cm, respectively; see Fig 3D for experimental details. The figure shows the fraction of bacteria that remain bound at the end of each 180-second flow step, determined as the difference between bacteria count at the start and end of each flow step. The error bars give 95% confidence intervals of 12 replicates.
(TIF)

**S4 Fig. Dr+ *E. coli* pre-bound to type IV collagen are resistant to detachment by flow with high shear stress.** (A), (B) and (C): the flow detachment experiment using dishes coated with 20, 2 or 0.2 μg ml$^{-1}$ human type IV collagen; see Fig 4C for experimental details. The figure shows the fraction of bacteria that remain bound at the end of each 192-second flow step, determined as the difference between bacteria count at the start and end of each flow step. The error bars give 95% confidence intervals of 12 replicates.
(TIF)

**S5 Fig. Statistics of formation of an increased adherence zone of Dr+ bacteria behind a stationary T24 budding host cell.** Dr+ *E. coli* were washed through a flow chamber with an overgrown layer of T24 cells with 24 ± 8 budding cells in the field of view, at a shear stress of 0.42 pN μm$^{-2}$ for 20 minutes. (A): increased adherence zones of Dr+ bacteria, characterized by well-defined boundaries. (B): zone area was determined using 'Measure' command, and the final number of bound bacteria in a given zone was determined using the 'Count and Measure' package of Olympus cellSens software. Time of zone formation was counted from the moment of attachment of the first until the attachment of the last bacteria. In panels (A) and (B) white and red arrows mark the positions of the budding cell and direction of flow, respectively. Bars correspond to 50 μm. Table (C) represents zone areas (5% accuracy), the number of accumulated bacteria (3% accuracy) and time of zone formation for 11 well-defined adherence zones. From these data, the rate of bacteria accumulation in zone per second and per second per mm$^2$ was calculated with an error of 3% and 6%, respectively. *—denotes data for adherence zone presented in panels (A) and (B).
(TIF)

**S1 Video. Representative time lapse videos of accumulation of Dr+ *E. coli* on the bladder transitional T24 cells under shear stress.** Experimental conditions are identical as in Fig 2. The video file shows the entire 20 minute experiments accelerated 30-fold.
(MP4)

**S2 Video. Representative time lapse videos of control experiments.** Attachment of Dr+ *E. coli* to: (A) a plastic tissue culture dish, passivated with flowing medium at static conditions at room temperature for 10 minutes; (B) bladder carcinoma T24 cells incubated with anti-DAF antibodies (as described in Methods). Attachment of Dr- *E. coli* to: (C) human type IV collagen at a concentration of 20 μg ml$^{-1}$ and (D) bladder carcinoma T24 cells. In all experiments, bacteria at $OD_{600}$ = 0.1 were washed through the flow chamber at shear stress of 0.01 pN μm$^{-2}$ for 20 minutes. The video file shows the entire 20 minute experiments accelerated 100-fold.
(MP4)

**S3 Video. Representative time lapse videos of detachment of Dr+ *E. coli* from the surface of bladder T24 cells under increasing shear stress.** Experimental conditions are identical as in Fig 2D. The beginning of the video shows bacterial accumulation prior to stepwise increase in flow. The video file shows the entire 28 minute experiment without the first 4 minutes of flow at shear stress of 0.28 pN μm$^{-2}$, accelerated 30-fold.
(MP4)

**S4 Video. Representative time lapse videos of detachment of Dr+ *E. coli* from the surface of bladder T24 cells under increasing shear stress in 2 mM Cm.** Experimental conditions are identical as in Fig 3D. The beginning of the video shows bacterial accumulation prior to stepwise increase in flow. The video file shows the 24 minute experiment without the first 3 minutes of flow at shear stress of 0.06 pN μm$^{-2}$, accelerated 30-fold.
(MP4)

**S5 Video. Representative time lapse videos of accumulation of Dr+ *E. coli* to flow chamber dishes coated with human type IV collagen under shear stress of 0.01 pN μm$^{-2}$.** Dishes were incubated with 20, 2 and 0.2 μg ml$^{-1}$ collagen solutions. Experimental conditions are identical as in Fig 4. The video file shows the entire 16 minute experiments accelerated 20-fold.
(MP4)

**S6 Video. Representative time lapse videos of accumulation of Dr+ *E. coli* to flow chamber dishes coated with human type IV collagen under shear stress of 0.06 pN μm$^{-2}$.** Description identical to Video 5.
(MP4)

**S7 Video. Representative time lapse videos of detachment of Dr+ *E. coli* from the flow chamber dishes coated with human type IV collagen under increasing shear stress.** The dish was incubated with a 0.2 μg ml$^{-1}$ collagen solution. The beginning of the video shows bacterial accumulation prior to stepwise increase in flow. The video file shows the entire 16 minute experiment accelerated 30-fold; changes in shear stress follow the description in Fig 4C.
(MP4)

**S8 Video. Representative time-lapse videos showing the layer of overgrown T24 cells with budding cells.** In the field of view a flow chamber is shown with visible single or multiple host cells budding from the confluent monolayer. Subtle forward and reverse movements of the flow medium were induced to observe slight movement of the budding cells. The video is accelerated 30-fold.
(MP4)

**S9 Video. Representative time lapse videos showing the influence of structural defects of the T24 cell layer on the accumulation of Dr+ and Dr- *E. coli* at elevated shear stress.** Washing of (A) Dr- and (B) Dr+ bacteria through the flow chamber with an overgrown layer of T24 cells characterized by occurrence of budding cells. Description identical as in Fig 6B and 6D. Washing of (C) Dr- and (D) Dr+ bacteria trough the flow chamber with an overgrown layer of T24 cells characterized by occurrence of detached rolling host cells. Description identical as in Fig 7D and 7F. The video file shows the entire 20 minute experiments accelerated 20-fold.
(MP4)

**S10 Video. Sample recordings of budding T24 cell at shear stress of 0.42 pN $\mu m^{-2}$.** In the negative control run (first 6 seconds), four different budding cells are marked to exemplify the diverse morphologies observed in the experiment. In the run with adhesion-competent bacteria (from 6 seconds on), continuous formation of an adherence zone can be observed. Video accelerated 3-fold.
(MP4)

**S11 Video. Sample recordings of rolling T24 cell at shear stress of 1.15 pN $\mu m^{-2}$.** In the negative control run (first 6 seconds), the highlighted cell is quickly removed by the flow and disappears from the focal surface. In the run with adhesion-competent bacteria, the cell slowly rolls on the cell line surface, continuously accumulating and depositing bacteria as it traverses the field of view. Video accelerated 3-fold.
(MP4)

## Author Contributions

**Conceptualization:** Beata Zalewska-Piątek, Tomasz Lipniacki, Sławomir Błoński, Miłosz Wieczór, Piotr Bruździak, Rafał Piątek.

**Investigation:** Beata Zalewska-Piątek, Marcin Olszewski, Tomasz Lipniacki, Sławomir Błoński, Anna Skwarska, Rafał Piątek.

**Methodology:** Beata Zalewska-Piątek, Marcin Olszewski, Tomasz Lipniacki, Sławomir Błoński, Piotr Bruździak, Anna Skwarska.

**Project administration:** Rafał Piątek.

**Supervision:** Rafał Piątek.

**Writing – original draft:** Beata Zalewska-Piątek, Marcin Olszewski, Tomasz Lipniacki, Sławomir Błoński, Miłosz Wieczór, Piotr Bruździak, Anna Skwarska, Bogdan Nowicki, Stella Nowicki, Rafał Piątek.

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
