## [Decision Letter · Decision Letter 0]

26 Jul 2019

Dear Dr. Piatek,

Thank you very much for submitting your manuscript "A shear stress micromodel of urinary tract infection by the Escherichia coli producing Dr adhesin" (PPATHOGENS-D-19-01145) for review by PLOS Pathogens. Your manuscript was fully evaluated at the editorial level and by independent peer reviewers. Reviewers appreciated the originality of the study and the potential broad interest. Importantly, however, reviewers also pointed out that experimental design and analysis of the key experiments is sometimes not sufficiently thorough. Reviewers make a number of suggestions to include further controls, quantifications of the results and further analysis that should be included in the revised manuscript to be considered for publication. We cannot, of course, promise publication at that time.

We therefore ask you to modify the manuscript according to the review recommendations before we can consider your manuscript for acceptance. Your revisions should address the specific points made by each reviewer.

(1) A letter containing a detailed list of your responses to the review comments and a description of the changes you have made in the manuscript. Please note while forming your response, if your article is accepted, you may have the opportunity to make the peer review history publicly available. The record will include editor decision letters (with reviews) and your responses to reviewer comments. If eligible, we will contact you to opt in or out.

(2) Two versions of the manuscript: one with either highlights or tracked changes denoting where the text has been changed; the other a clean version (uploaded as the manuscript file).

Additionally, to enhance the reproducibility of your results, PLOS recommends that you deposit your laboratory protocols in protocols.io, where a protocol can be assigned its own identifier (DOI) such that it can be cited independently in the future. For instructions see http://journals.plos.org/plospathogens/s/submission-guidelines#loc-materials-and-methods

We hope to receive your revised manuscript within 60 days. If you anticipate any delay in its return, we ask that you let us know the expected resubmission date by replying to this email. Revised manuscripts received beyond 60 days may require evaluation and peer review similar to that applied to newly submitted manuscripts.

[LINK]

Sincerely,

Guillaume Dumenil

Guest Editor

PLOS Pathogens

Xavier Nassif

Section Editor

PLOS Pathogens

Kasturi Haldar

Editor-in-Chief

PLOS Pathogens

orcid.org/0000-0001-5065-158X

Grant McFadden

Editor-in-Chief

PLOS Pathogens

orcid.org/0000-0002-2556-3526

Reviewers appreciated the originality of the study and the potential broad interest. Importantly however reviewers also point out that experimental design and analysis of the key experiments is sometimes not sufficiently thorough. Reviewers make a number of suggestions to include further controls, quantifications of the results and further analysis that should be included in the revised manuscript to be considered for publication.

Reviewer's Responses to Questions

**Part I - Summary**

Reviewer #1: The observations in this study about the role of budding cells in the adhesion and spreading of Dr+ E. coli are quite novel and highly relevant to pathogenesis in a way that may extend beyond the novel system. However, the data presented lack quantitative analysis, demonstration of reproducibility, and statistical analysis, all of which are required to support the conclusions drawn. The fluid mechanics simulations are an excellent addition that provide important insights into interpretation of this data. The data on the effect of shear stress on bacterial adhesion is rigorously presented with minor exceptions, and is critical to interpretation of the budding/rolling cell data. An additional concern is that the manuscript does not satisfactorily justify or explain to what degree the budding cells provide a physiologically relevant model for exfoliation during infection. In summary, the strengths of this study are the novelty and significance of the key findings regarding the budding and detached cells, and the variety of experiments that together provide significant insights into mechanisms that underlie the key findings. The weaknesses are the lack of rigorous experimental design and analysis of the key experiments, and the lack of clarity about the appropriateness of the budding cell model.

Reviewer #2: Zalewska-Piątek et al. characterize E. coli Dr bonds with host cells. They conclude that Dr bonds are slip bonds, occur between the DAF receptor and collagen, that epithelial architecture affects adhesion of cells through Dr bonds, and that rolling host cells increases the spread of bacterial adhesion to host cell surfaces. The topic is interesting and the conclusions are mostly supported by the data. However, in some cases sufficient data is not shown.

Reviewer #3: Manuscript by Zalewska-Piątek et al. describes several phenomena of interaction between Dr-adhesin expressing E. coli and T24 urothelial cell monolayer under various flow conditions. The main observations include shear-inhibited slip-bond adhesion of E. coli with various modes of attachment/detachment, bacterial focal adhesion behind the budding cells and bacteria-mediated rolling of detached epithelial cells over the cell monolayer.

Overall the experiments are done well and lots of in-depth video analysis and simulations had been performed. The slip-bond mechanism of Dr-mediated adhesion is shown in a convincing manner. The most intriguing is the binding of E. coli behind the budding cells, opposite from the flow direction. Such mode of adhesion, probably due to the bacteria flow-mediated trapping, could explain certain phenomena seen in other adhesion models, highlighting the need to consider microfluidic parameters of the environment. Another intriguing observation is that bacteria on sloughed-off epithelial cell could mediate the cell rolling and use it as a vehicle to seed bacteria to new cells by relocating from one cell to another.

**Part II – Major Issues: Key Experiments Required for Acceptance**

Reviewer #1: No new experiments are required, but two experiments should be modified to analyze the data to show that the findings are reproducible and statistically significant. Specifically, the following observations were only supported by images that don’t address experimental rigor: (Adhesion to cultured cells is often spotty, and a few images can’t address whether the differences are due to the budding cells rather than random variation on the surface, or even random variation between two plates of cells.)

1. Line 290: “Budding of T24 cells induces efficient adherence of Dr+ E. coli at elevated shear stress”.

2. Line 339: “Rolling of T24 cells facilitates the spread of Dr+ E. coli at elevated shear stress”.

Reviewer #2: Major issues:

1. Lines 196-199: The authors claim that the results are identical for constant and peristaltic-like flow but show no data. The authors should show data to support these claims.

2. Lines 495-498: One of the main conclusions of the manuscript is that the Dr bonds are slip bonds and not catch bonds. This conclusion is based heavily on the results of Fig. 2. However, no slip or catch bonds are shown for reference. Can the authors show as controls how known catch bonds and slip bonds would appear in their system?

3. Fig. 3: The authors incubate the bacteria with above-MIC concentrations of chloramphenicol for 20 minutes, which would completely inhibit bacterial protein synthesis. This effect could interfere with adhesion and have little to do with inhibition of receptor binding. The authors should differentiate between translational inhibition and receptor binding, such as using a chloramphenicol-resistant strain.

4. Lines 236-238: The authors make a clinical claim about therapeutics but the system being studied is very different from a clinical setting. It remains to be determined if the result would apply.

5. The bonding mechanism for Dr to collagen is unclear. Does the bond depend on the slip mechanism presented in Figs. 2-3? The use of the chloramphenicol could address this. The use of anti-DAF antibody would serve as a good negative control, as this should not affect the Dr-collagen bond.

6. Lines 276-278: The data does not support that bacteria roll on the cell surface. It is possible that they merely slide along the surface.

7. Fig. 6: The attachment of cells behind budding cells is unclear. Showing phase contrast images may improve the interpretation.

8. Lines 334-336: It is unclear how the authors determined the experimental attachment rate from the data. The authors should show how this was determined and give details of the statistics of the measurement.

9. Lines 336-338: It is unclear how this statement is relevant to the results of the model.

10. Lines 352-353: The transfer of bacteria onto host surfaces by rolling cells is interesting. However, the analysis of these results is not convincing.

11. Lines 360-361: The authors cite a graphic as evidence.

Reviewer #3: none

**Part III – Minor Issues: Editorial and Data Presentation Modifications**

Reviewer #1: 1. Figure 1: CEA abbreviation needs to be defined before it is used. It is defined now in the methods, but this will follow the figure in the formatted manuscript.

2. Figure 2B and 2C need reformatting. – the insets are too small to read. In 2B, the scaling of the y-axis to start at -40 makes the data visually misleading for the main plot (it likes like it drops by ½, but it actually drops much more. Yes, any such misinterpretations by the reader are corrected by the text itself, but the visual representation is distracting). Also, these two panels are redundant; any data of interest can be included as counts per 12 mins, or counts per minute, and put in one graph. Either way, please include a legend in the figure itself and use different symbols so it can be read if viewed in grayscale or by a colorblind reader.

3. Figure 2D – start graph at 0 to avoid misleading visualization of the data, and use larger text for the shear stress. Also, this is a representative graph. The experiment should be repeated, and the statistics plotted. Perhaps plot fraction left at end of shear stress on y vs shear stress on x? Also, the “data not shown” on the peristalsis question should be shown (with the statistics of multiple runs) or left out.

4. Figure 3 – same formatting issues and lack of demonstration of reproducibility as in figure 2.

5. Some negative controls are included only as supplementary videos. Since they are critical to the conclusions, they instead should be included in the main paper, ideally as quantified data with replicates and statistics. (e.g. plot number bound in the two negative controls in figure 2B/C).

6. The manuscript does not satisfactorily justify or explain to what degree the budding cells provide a physiologically relevant model for exfoliation during infection. Specifically, what is the relationship, if any, between budding and exfoliation? This should be addressed before the budding experiments are described, and if budding is not actually involved in exfoliation, the strengths and limitations of this model should be addressed in the discussion. Specifically, do exfoliating cells round up or flake off, or is this currently not known?

7. Figure 6 is confusing; I had to read it five times to find what was different between the panels (Dr- vs Dr + and confluent vs budding). Also, is the flow direction vertical in B-D?

8. It is stated that the Dr+ E.coli are what mediates epithelial cell rolling (page 15), but this is not clear without a negative control showing that budding cells don’t roll without Dr+ E. coli. This statement should be qualified to address the low level of certainty, or removed.

Reviewer #2: Minor comments:

1. Line 152: The authors should give the dimensions of the flow chamber here or in the materials.

2. Line 166, throughout: “Thanks to” should be changed to a phrase that is more precise such as “Due to”.

3. Fig. 6-7: The authors need to indicate the direction of flow.

4. Line 314: “is also smal” should be “is also small”?

Reviewer #3: I think that the collagen and chloramphenicol experiments do not add much to the study and could be eliminated to make the overly long manuscript more focused. Also, lots of the discussion is highly speculative, especially about the low-symptomatic nature of UTIs caused by Dr-fimbriated E. coli, which is very uncertain.

PLOS authors have the option to publish the peer review history of their article (what does this mean?). If published, this will include your full peer review and any attached files.

Reviewer #1: Yes: Wendy E. Thomas

Reviewer #2: No

Reviewer #3: No

---

## [Decision Letter · Decision Letter 1]

28 Nov 2019

Dear Dr Piatek,

We are pleased to inform that your manuscript, "A shear stress micromodel of urinary tract infection by the Escherichia coli producing Dr adhesin", has been editorially accepted for publication at PLOS Pathogens. The three reviewers appreciated the efforts to address their recommendations and the quality of the final manuscript. Please note that Reviewer 1 pointed two minor points that need to be modified.

1.     In figure 3, Cm concentration should be shown in micromolar µM, not micrometer �m.  

2.     Most figures are now fine, but figure 4 has text so small it can’t be read at 100% size in all three panels.

Before your manuscript can be formally accepted and sent to production, you will need to complete our formatting changes, which you will receive by email within a week. Please note that your manuscript will not be scheduled for publication until you have made the required changes.

IMPORTANT NOTES

(1) Please note, once your paper is accepted, an uncorrected proof of your manuscript will be published online ahead of the final version, unless you’ve already opted out via the online submission form. If, for any reason, you do not want an earlier version of your manuscript published online or are unsure if you have already indicated as such, please let the journal staff know immediately at plospathogens@plos.org.

(2) Copyediting and Proofreading: The corresponding author will receive a typeset proof for review, to ensure errors have not been introduced during production. Please review the PDF proof of your manuscript carefully, as this is the last chance to correct any errors. Please note that major changes, or those which affect the scientific understanding of the work, will likely cause delays to the publication date of your manuscript. 

(3) Appropriate Figure Files: Please remove all name and figure # text from your figure files. Please also take this time to check that your figures are of high resolution, which will improve the readbility of your figures and help expedite your manuscript's publication. Please note that figures must have been originally created at 300dpi or higher. Do not manually increase the resolution of your files. For instructions on how to properly obtain high quality images, please review our Figure Guidelines, with examples at: http://journals.plos.org/plospathogens/s/figures.

(4) Striking Image: Please upload a striking still image to accompany your article if one is available (you can include a new image or an existing one from within your manuscript). Should your paper be accepted, this image will be considered for our monthly issue image and may also appear on our website to feature your article. Please upload this as a separate file, selecting "striking image" as the file type upon upload. Please also include a separate "Other" file with a caption, including credits and any potential copyright information. Please do not include the caption in the main article file. If your image is from someone other than yourself, please ensure that the artist has read and agreed to the terms and conditions of the Creative Commons Attribution License at http://journals.plos.org/plospathogens/s/content-license. Please note that PLOS cannot publish copyrighted images.

(5) Press Release or Related Media: If your institution or institutions have a press office, please notify them about your upcoming paper at this point, to enable them to help maximize its impact. If they will be preparing press materials for this manuscript, please inform our press team in advance at plospathogens@plos.org as soon as possible. We ask that you contact us within one week to plan ahead of our fast Production schedule. If you need to know your paper's publication date for related media purposes, you must coordinate with our press team, and your manuscript will remain under a strict press embargo until the publication date and time. This means an early version of your manuscript will not be published ahead of your final version. 

(6)  PLOS requires an ORCID iD for all corresponding authors on papers submitted after December 6th, 2016. Please ensure that you have an ORCID iD and that it is validated in Editorial Manager.  To do this, go to ‘Update my Information’ (in the upper left-hand corner of the main menu), and click on the Fetch/Validate link next to the ORCID field.  This will take you to the ORCID site and allow you to create a new iD or authenticate a pre-existing iD in Editorial Manager

(7) Update your Profile Information: Now that your manuscript has been provisionally accepted, please log into Editorial Manager and update your profile, if needed. Go to https://www.editorialmanager.com/ppathogens, log in, and click on the "Update My Information" link at the top of the page. Please update your user information to ensure an efficient production and billing process. 

(8) LaTeX users only: Our staff will ask you to upload a TEX file in addition to the PDF before the paper can be sent to typesetting, so please carefully review our Latex Guidelines http://journals.plos.org/plospathogens/s/latex in the meantime.

(9) If you have associated protocols in protocols.io, please ensure that you make them public before publication to guarantee immediate access to the methodological details.

Best regards,

Guillaume Dumenil

Guest Editor

PLOS Pathogens

Xavier Nassif

Section Editor

PLOS Pathogens

Kasturi Haldar

Editor-in-Chief

PLOS Pathogens

orcid.org/0000-0001-5065-158X

Grant McFadden

Editor-in-Chief

PLOS Pathogens

orcid.org/0000-0002-2556-3526

Dear Dr Piatek,

Thank you very much for providing the requested changes in the manuscript. Reviewers have appreciated the efforts made to respond to their comments and all agree that the manuscript is now fit for publication in PLOS Pathogens. Please note that Reviewer 1 has indicated two minor changes that will need to be made before publication.

Best Regards,

Guillaume Duménil

Reviewer Comments (if any, and for reference):

Reviewer's Responses to Questions

Part I - Summary

Reviewer #1: The manuscript was greatly improved by the changes. The added clarification of significance and the rigor of the experiments now fully support the conclusions that I found most exciting in the first version. There are just some minor corrections that should be addressed for readability.

Reviewer #2: The authors have addressed my comments.

Reviewer #3: I am satisfied with the responses to my critique. And considering the major modifications, the manuscript became much stronger.

Part II – Major Issues: Key Experiments Required for Acceptance

Please use this section to detail the key new experiments or modifications of existing experiments that should be 

absolutely

 required to validate study conclusions.

Reviewer #1: (No Response)

Reviewer #2: (No Response)

Reviewer #3: none

Part III – Minor Issues: Editorial and Data Presentation Modifications

Reviewer #1: Minor comments:

1. In figure 3, Cm concentration should be shown in micromolar µM, not micrometer �m.

2. Most figures are now fine, but figure 4 has text so small it can’t be read at 100% size in all three panels.

Reviewer #2: (No Response)

Reviewer #3: none

PLOS authors have the option to publish the peer review history of their article (what does this mean?). If published, this will include your full peer review and any attached files.

Do you want your identity to be public for this peer review?

 For information about this choice, including consent withdrawal, please see our Privacy Policy.

Reviewer #1: No

Reviewer #2: No

Reviewer #3: Yes: Evgeni Sokurenko

---

## [Editor Report · Acceptance letter]

31 Dec 2019

Dear Dr. Piatek,

We are delighted to inform you that your manuscript, "A shear stress micromodel of urinary tract infection by the Escherichia coli producing Dr adhesin," has been formally accepted for publication in PLOS Pathogens.

Best regards,

Kasturi Haldar

Editor-in-Chief

PLOS Pathogens

orcid.org/0000-0001-5065-158X

Grant McFadden

Editor-in-Chief

PLOS Pathogens

orcid.org/0000-0002-2556-3526